# Structural basis for the activation of acid ceramidase

Ahmad Gebai[1], Alexei Gorelik [1], Zixian Li[1], Katalin Illes[1] & Bhushan Nagar [1]

Acid ceramidase (aCDase, *ASAH1*) hydrolyzes lysosomal membrane ceramide into sphingosine, the backbone of all sphingolipids, to regulate many cellular processes. Abnormal function of aCDase leads to Farber disease, spinal muscular atrophy with progressive myoclonic epilepsy, and is associated with Alzheimer's, diabetes, and cancer. Here, we present crystal structures of mammalian aCDases in both proenzyme and autocleaved forms. In the proenzyme, the catalytic center is buried and protected from solvent. Autocleavage triggers a conformational change exposing a hydrophobic channel leading to the active site. Substrate modeling suggests distinct catalytic mechanisms for substrate hydrolysis versus autocleavage. A hydrophobic surface surrounding the substrate binding channel appears to be a site of membrane attachment where the enzyme accepts substrates facilitated by the accessory protein, saposin-D. Structural mapping of disease mutations reveals that most would destabilize the protein fold. These results will inform the rational design of aCDase inhibitors and recombinant aCDase for disease therapeutics.

[1] Department of Biochemistry and Groupe de Recherche Axé sur la Structure des Protéines, McGill University, Montreal, QC H3G 0B1, Canada. These authors contributed equally: Ahmad Gebai, Alexei Gorelik. Correspondence and requests for materials should be addressed to B.N. (email: bhushan.nagar@mcgill.ca)

Sphingolipids are a class of diversely modified bioactive lipids in cellular membranes that have crucial roles in biological processes including proliferation, apoptosis, autophagy, metabolism, inflammation, and stress responses[1]. They are also responsible for forming membrane microdomains termed lipid rafts, which are important for regulating signal transduction pathways through clustering of receptor proteins[2]. Ceramide is the central nexus in sphingolipid metabolism where it can be modified with head groups to form complex sphingolipids such as ceramide-1-phosphate, sphingomyelin, and glycosphingolipids, or broken down into sphingosine and free fatty acid[3]. Importantly, sphingosine is the backbone for all mammalian sphingolipids and is generated exclusively through the breakdown of ceramides by ceramidases[4].

Humans possess five ceramidases, classified by their pH optimum (alkaline, neutral, and acid), cellular localization, primary structure, and function[5]. Neutral ceramidase (nCDase) is important in the catabolism and digestion of dietary sphingolipids, and in regulating the level of sphingolipid metabolites in the intestinal tract[6]. Alkaline ceramidases, of which there are three paralogs (ACER1, ACER2, and ACER3), have roles in cell differentiation[7], DNA damage-induced cell death[8], and cell proliferation[9]. Acid ceramidase (aCDase; *ASAH1*) is the best studied member of the family and is prominent for its involvement in Farber disease (FD), a genetic disorder in humans[4].

aCDase is ubiquitously expressed and found in the lysosome where it normally regulates sphingolipid metabolism through intralysosomal membrane turnover[10]. Because of this essential housekeeping role, loss-of-function mutations in aCDase can cause two severe diseases: FD and spinal muscular atrophy with progressive myoclonic epilepsy (SMA-PME)[11, 12]. FD is a lysosomal storage disease caused by the accumulation of ceramide, leading to joint pain and deformation, throat abnormalities, and early death[13]. SMA-PME is an autosomal recessive neurodegenerative disease causing atrophy from the loss of motor neurons, accompanied by myoclonic epilepsy[12, 14]. Increased expression of aCDase or accumulation of ceramides in cells has also been linked to diseases such as melanoma[15, 16], head and neck cancers[17, 18], acute myeloid leukemia (AML)[19], Alzheimer's[20], and type-2 diabetes[21]. Thus, aCDase is a critical target for inhibitor development in diseases involving excess hydrolysis of ceramide and accumulation of sphingosine-1-phosphate, and for enzyme

replacement therapy in diseases caused by dysfunctional aCDase. Indeed, aCDase was suggested as a drug target for treatment of AML and pediatric brain tumors[22]. Moreover, injection of lentivectors encoding wild-type human-aCDase rescued the phenotype observed in mice encoding a faulty enzyme[23] and there is ongoing pharmaceutical interest in developing recombinant aCDase for FD and SMA-PME[24]. A clearer molecular understanding of aCDase function would facilitate the development of improved therapeutics.

aCDase is a glycosylated 50 kDa enzyme belonging to the N-terminal nucleophile (Ntn) superfamily of hydrolases[25, 26]. These proteins are synthesized as inactive proenzymes that are activated through autocleavage of an internal peptide bond[27]. The autocleavage of aCDase occurs at the peptide bond preceding Cys 143, rendering a mature heterodimeric enzyme comprised of a 13 kDa α-subunit and 30 kDa β-subunit[28, 29]. A catalytic mechanism for aCDase autocleavage was proposed based on a biochemical analysis[29], involving activation of Cys 143 (or Ser/Thr in other amidases) by a general base, formation of an acyl-intermediate stabilized by an oxyanion hole, and resolution of the complex by a water nucleophile[30]. Mature aCDase is thought to use the same Cys 143 nucleophile—the newly generated N terminus of the β-subunit—for hydrolysis of ceramide substrates. How autocleavage enables binding and activity on ceramide, and the details of its catalytic mechanism, are unknown.

An additional layer of complexity that distinguishes aCDase from other Ntn-hydrolases is that for optimal activity, it requires the protein saposin-D[31]. Saposins are small lysosomal "helper" proteins that can assume a globular closed conformation or an open V-shaped form capable of lipid interactions[32]. In some cases, they act in isolation by extracting and solubilizing lipids from membranes via their open configuration for subsequent breakdown by hydrolytic enzymes[33]. Alternatively, they can establish complexes with an enzyme for delivery of lipid substrate and/or allosteric activation of enzymatic activity[34]. Studies suggest that saposin-D is likely to be involved in membrane disruption to facilitate ceramide access by aCDase[35, 36]. Recently, two groups determined the structure of saposin-D in a closed, lipid-free form, describing a buried hydrophobic surface that could be involved in lipid-binding[36, 37].

Although aCDase has been extensively studied since its initial discovery, structural data are still missing for a complete

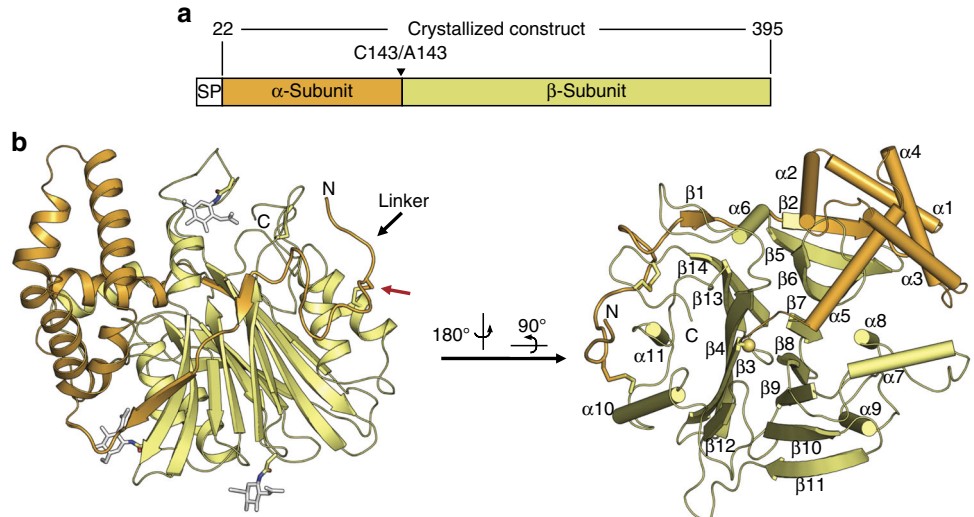

**Fig. 1** Structural overview of aCDase. **a** Domain organization of aCDase (SP, signal peptide). **b** Left, structure of the inactive form of recombinant aCDase from the naked mole rat, colored as shown in **a**. The location of the disulfide bridge linking the α- and β-subunits is indicated by a red arrow. The glycans produced from insect cell expression are displayed as white sticks. Right, structure of aCDase in a different view with secondary structure elements labeled

**Table 1 Crystallographic data collection and refinement statistics**

|  | aCDase from nmr, inactive | aCDase from cmw, inactive | Human aCDase, activated |
|---|---|---|---|
| PDB code | 5U81 | 5U84 | 5U7Z |
| Data collection |  |  |  |
| Space group | C 2 | C 2 | C 2 |
| Unit cell a, b, c (Å) | 160.77, 53.91, 48.80 | 118.82, 109.27, 79.54 | 153.67, 68.31, 97.81 |
| Unit cell α, β, γ (°) | 90, 104.41, 90 | 90, 102.97, 90 | 90, 120.72, 90 |
| Wavelength (Å) | 0.98 | 1.77 | 0.98 |
| Resolution range (Å) | 100–1.40 (1.45–1.40) | 100–2.34 (2.42–2.34) | 50–2.50 (2.59–2.50) |
| Total reflections | 292,372 (24,074) | 278,809 (14,715) | 110,914 (8327) |
| Unique reflections | 78,833 (7121) | 40,925 (3977) | 30,295 (2834) |
| Multiplicity | 3.7 (3.4) | 6.8 (3.7) | 3.7 (2.9) |
| Completeness (%) | 99 (90) | 99 (97) | 99 (94) |
| $R_{meas}$ (%) | 5 (104) | 14 (172) | 12 (148) |
| $R_{pim}$ (%) | 3 (53) | 6 (62) | 8 (76) |
| Mean $I/\sigma(I)$ | 27.8 (1.6) | 14.4 (1.3) | 10.8 (0.8) |
| $CC_{1/2}$ | (0.68) | (0.50) | (0.34) |
| Wilson B factor (Å$^2$) | 14.5 | 32.0 | 36.1 |
| Refinement |  |  |  |
| Protein copies per ASU | 1 | 2 | 2 |
| Resolution range (Å) | 36.57–1.40 (1.45–1.40) | 44.66–2.34 (2.42–2.34) | 48.39–2.50 (2.59–2.50) |
| Reflections used | 69,315 (4018) | 39,726 (2798) | 26,667 (1404) |
| Reflections for $R_{free}$ | 3490 (209) | 1922 (149) | 1,335 (71) |
| $R_{work}$ (%) | 12.2 (15.8) | 20.0 (31.3) | 23.4 (31.4) |
| $R_{free}$ (%) | 15.8 (24.6) | 23.1 (31.3) | 27.3 (31.2) |
| Non-hydrogen atoms | 3559 | 6113 | 6128 |
| Protein | 3053 | 5665 | 5882 |
| Glycans, ligands, ions | 203 | 314 | 213 |
| Water | 303 | 134 | 33 |
| Average B factor (Å$^2$) | 24.7 | 44.7 | 37.4 |
| Protein | 22.4 | 44.3 | 36.7 |
| Glycans, ligands, ions | 37.6 | 54.4 | 59.8 |
| Water | 39.5 | 38.6 | 33.4 |
| RMSD bond lengths (Å) | 0.008 | 0.003 | 0.003 |
| RMSD bond angles (°) | 1.17 | 0.73 | 0.68 |
| Ramachandran favored (%) | 97.6 | 98.0 | 97.1 |
| Ramachandran allowed (%) | 2.4 | 2.0 | 2.9 |
| Ramachandran outliers (%) | 0 | 0 | 0 |
| Rotamer outliers (%) | 0.9 | 1.1 | 1.5 |
| Clashscore | 2.8 | 5.6 | 4.7 |

Values in parentheses are for the highest-resolution shell
aCDase, acid ceramidase; ASU, asymmetric unit; cmw, common minke whale; nmr, naked mole rat; RMSD, root-mean-square deviation

understanding of its function. Here, we present crystal structures of both the proenzyme and mature (cleaved) forms of aCDase. The structures display a fold similar to that of Ntn-hydrolases. Comparison of the proenzyme to the active form reveals that a conformational rearrangement of the cleaved region and the preceding α-helix uncovers the active site for entry of ceramide substrates. The structures also provide insight into the catalytic mechanisms for autocleavage and substrate hydrolysis, which appear to use distinct general bases and oxyanion holes. Strikingly, an expansive hydrophobic surface surrounds the uncovered active site in the mature enzyme, likely havng a role in membrane interactions. Finally, mapping of disease mutations provides structural insight into their malignant phenotypes.

## Results

**Overall structure**. We determined high-resolution crystal structures of several aCDases from various mammalian species and distinct activation states (Fig. 1a and Table 1). Two structures are of the full-length proenzyme before autocleavage, from naked mole rat (nmr, 83% identity to human) and common minke whale (cmw, 85% identity). The autocleavage reaction was prevented in these structures by mutating the catalytic cysteine residue to alanine (C143A). A third structure is of the heterodimeric activated state from human, which was allowed to fully

autocleave before crystallization. All proteins were recombinantly produced in insect cells.

Both the proenzyme and active states are similar with regards to overall structure and subunit disposition. The α-subunit consists of a globular helical domain formed by five α-helices preceded by a 36-residue linker region, whereas the β-subunit contains two central anti-parallel β-sheets flanked on either side by a total of six α-helices (Fig. 1b). The helical region of the α-subunit abuts against one side of the β-subunit with helix-α5 reaching into the β-subunit up to the cleavage site (Fig. 1b, right). Meanwhile, the N-terminal linker of the α-subunit wraps around half the β-subunit as it inserts two short β-strands to complete the edges of both β-sheets. The protein contains six cysteines, four of which form two disulfide bonds. One disulfide stabilizes a turn in the β-subunit (C388/C392), whereas the other covalently latches the N-terminal end of the α-subunit linker to the β-subunit (C31/C340) (Fig. 1b, red arrow). Thus, the α- and β-subunits are intimately associated burying a total of 894 Å$^2$, maintaining a heterodimeric state even after autocleavage.

Although the overall structures of the proenzymes from nmr-aCDase and cmw-aCDase are similar (root-mean-square deviation (r.m.s.d.) = 0.73 Å), there are differences. In the α-subunit, the orientations and lengths of helices, particularly helices-α2 and -α3, are slightly different (Supplementary Fig. 1A). The β-

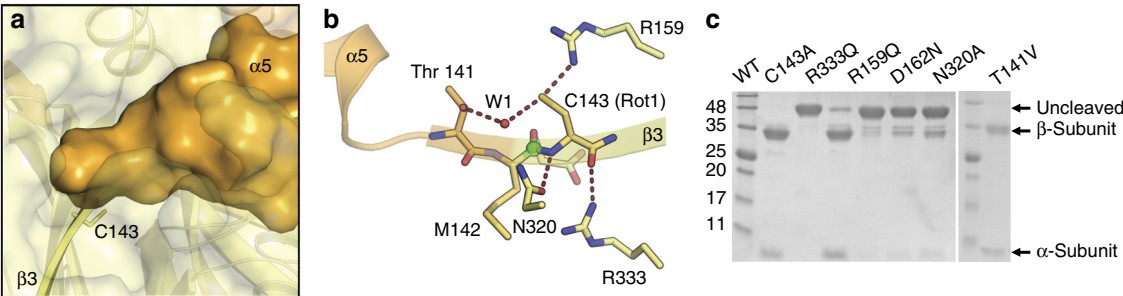

**Fig. 2** The inactive state of aCDase. **a** The presence of an intact junction (residues 140–142 shown in surface representation) preceding the active site nucleophile sequesters Cys 143 (modeled; yellow sticks) from solvent. **b** Active site of the proenzyme form of nmr-aCDase. The modeled rotamer (Rot1) of the active site Cys 143 is shown as yellow sticks. The carbonyl carbon attacked by the Cys 143 thiol side chain is shown as a green sphere. Dashed red lines indicate hydrogen bonds. An active site water (W1) is shown as a red sphere. **c** Reducing SDS-PAGE of the purified active site mutants of aCDase following the autocleavage reaction

subunits are essentially identical, but there is a significant deviation in the ~14-residue loop connecting strand-β4 to helix-α6 (L4-6; Supplementary Fig. 1B). In nmr-aCDase, L4-6 forms an extended structure that protrudes 'outward,' whereas in cmw-aCDase it flips toward the center of the protein and sits atop the junction between the subunits. Interestingly, the conformation of L4-6 in active human-aCDase is essentially the same as in nmr-aCDase (Supplementary Fig. 1B). The significance of this loop variation is unclear, because in cmw-aCDase it is involved in a crystal contact (Supplementary Fig. 1C). However, as described below, L4-6 does appear to have functional importance.

aCDase is N-glycosylated with both cmw-aCDase and nmr-aCDase having four glycosylation sites, whereas human-aCDase possesses an additional two sites. Only a subset of these sites could be modeled, presumably because some are flexible and could not be visualized in the electron density. Three specific glycosylation sites were shown to be required for autocleavage to occur[38] and the structure reveals that two of them form extensive bridging interactions between the α- and β-subunits in the proenzyme (Supplementary Fig. 1D, E).

The structure of aCDase is in accordance with the αββα-fold of members of the Ntn-hydrolase superfamily[30], as seen by its superposition with three representative Ntn-hydrolases: mouse phospholipase B-like protein 2, Acyl-coenzyme A:6-aminopenicillanic-acid-acyltransferase from *Penicillium chrysogenum*, and the penicillin acylase from *Lysinibacillus sphaericus* (Supplementary Fig. 2A–C). Although the proteins differ significantly in their primary sequences, the β-subunit of aCDase aligns well with the other hydrolases, and the active site is found at the same location. The main differences are observed in the α-subunit, where the number and orientation of α-helices vary. Certain Ntn-hydrolases lack an α-subunit as in penicillin acylase, but the core αββα-fold is conserved[39].

**The inactive proenzyme state.** Similar to other Ntn-hydrolases, aCDase is synthesized as an inactive proenzyme that requires proteolysis for activation[28]. This involves autocleavage by an internal nucleophile, usually a threonine, serine, or cysteine that subsequently becomes the N terminus of the resultant β-subunit and active site for substrate hydrolysis[30]. Previous biochemical studies on aCDase revealed that Cys 143 is responsible for breaking the peptide bond between it and the preceding residue (Ile 142 in humans), most likely by intramolecular cleavage[29].

In the proenzyme structure, the peptide bond between Cys 143 (Ala 143 in the inactivated structures) and residue 142 (methionine in nmr-aCDase and isoleucine in cmw-aCDase) is intact as shown by continuous electron density (Supplementary Fig. 3). Cys 143 is the third residue of the first β-strand of the

putative β-subunit and its side chain points toward the β-sheet stacked underneath. Residues 141 and 142 complete the β-strand, which leads into a tight turn preceded by the C-terminal helix-α5 of the putative α-subunit (Fig. 2a). The presence of an intact junction combined with the orientation of Cys 143 completely sequester it and the scissile peptide bond from the outside environment, confirming that autocleavage must occur intramolecularly.

**Catalytic mechanism of autocleavage.** A catalytic mechanism for autocleavage was proposed previously based on homology modeling with cholylglycine hydrolase (14% identity)[29]. Besides Cys 143, two additional residues were suggested to participate in catalysis, Asp 162 and Arg 159. Arg 159 was posited as the general base to promote nucleophilic attack by the Cys 143 thiol on the preceding peptide bond to form an internal thioester subsequently resolved by hydrolysis, leaving Cys 143 as the N terminus of the β-subunit.

In both proenzyme structures, the disposition of the active site residues agrees with the proposed mechanism, though there are differences. Since both inactive structures are of the C143A mutant, the position of the thiol was not experimentally determined, but we could model it based on energetically accessible cysteine rotamers and their corresponding steric clashes with the protein. In both inactive structures, only one rotamer satisfies these criteria (Rot1; Fig. 2b). Further rotation of the chi-angle causes clashes with a tightly bound water molecule (W1) that appears to be important for the catalytic mechanism.

Cysteines generally require deprotonation by a general base before nucleophilic attack. The structure reveals that a possible general base is Arg 159 as it is the only residue close enough to hydrogen bond with the modeled thiol. Accordingly, mutation of Arg 159 dramatically reduces autocleavage activity (Fig. 2c). The Cys 143 thiol also hydrogen bonds with W1, which in turn hydrogen bonds to Thr 141. Thr 141 is conserved across all species of aCDase (Supplementary Fig. 4) and W1 is found in both inactive structures determined here, suggesting that it probably plays an important role. In theory, W1 could also act as the general base, but it more likely is important in the resolution of the thioester intermediate, as it is positioned directly over the carbonyl carbon of the scissile peptide. Mutation of Thr 141 significantly reduced the autocleavage rate to 40% of wild type (Supplementary Fig. 5), although after 65 h the mutant was almost fully cleaved (Fig. 2c). This is likely because W1 also contacts other residues (Asn 320 and Glu 225) and mutation of Thr 141 presumably only weakens its binding.

Other residues in proximity to Cys 143 that could have a catalytic role are Arg 333, Asn 320, and Asp 162 (Fig. 2b and

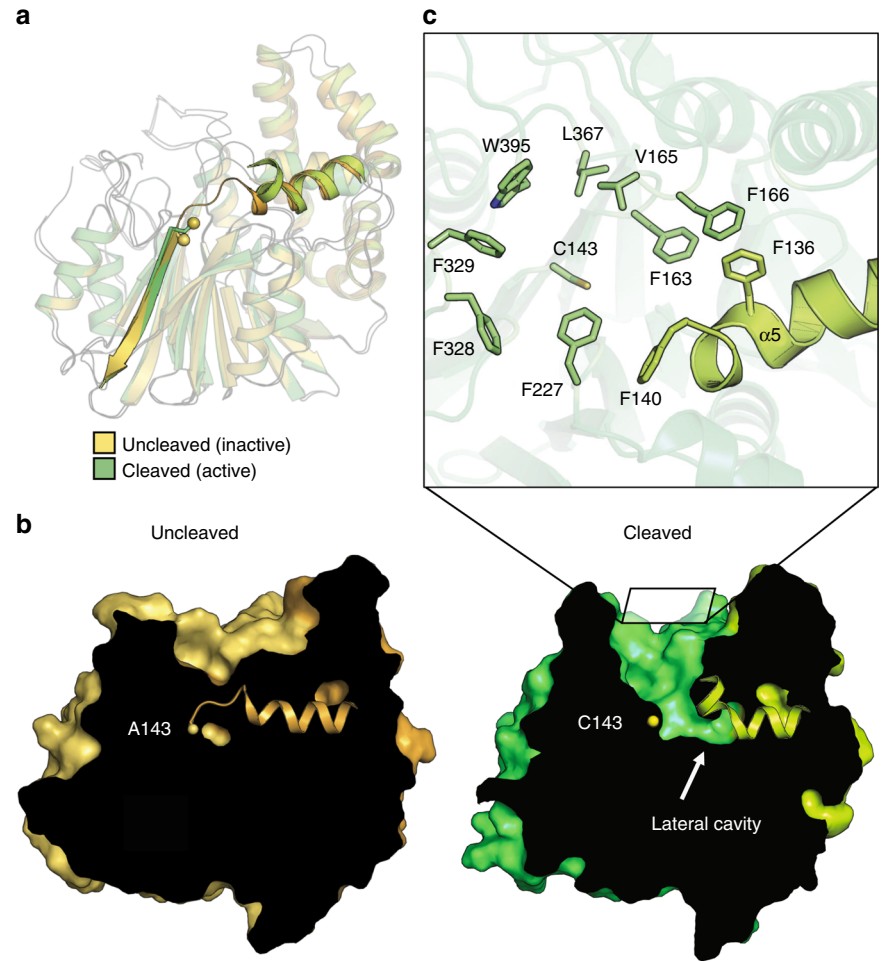

**Fig. 3** Activation of aCDase. **a** Comparison of the active (human) and inactive (nmr) forms of aCDase. The inactive nmr-aCDase is in yellow (β-subunit) and orange (α-subunit). The active enzyme is in green (β-subunit) and light green (α-subunit). The active site Cys 143 is represented by yellow spheres in both structures. Helix-α5 and strand-β3 are emphasized to show the difference before and after cleavage. **b** Cutaways of the inactive (left, yellow) and active forms (right, green) of aCDase. Helix-α5 is displayed as a ribbon. Ala 143/Cys 143 are shown as yellow spheres. The square above the cleaved enzyme indicates the direction of view in **c**. **c** The substrate binding channel exposed after autocleavage, viewed looking down into the channel. Hydrophobic residues in the channel that likely accommodate the ceramide lipid tails are shown as sticks (β-subunit, green sticks; α-subunit, light green sticks)

Supplementary Fig. 6). Arg 333 and Asn 320 appear to be important for orienting Cys 143 through hydrogen bonding to its backbone carbonyl and nitrogen, respectively. Asp 162 was shown to be important for catalysis from previous work[29]. It is located on an adjacent β-strand where its side chain is 4.3 Å from the backbone nitrogen of Cys 143 and hydrogen bonded with several surrounding residues including a weak ionic interaction with Arg 333. The backbone nitrogen of Asp 162 hydrogen bonds to the carbonyl oxygen of the scissile peptide, implicating it in stabilizing the transient oxyanion formed during catalysis. Mutation of Asn 320 and Asp 162 significantly reduced autocleavage activity, whereas mutation of Arg 333 had a lesser effect (Fig. 2c). The fact that the D162N mutation had such a strong effect is somewhat surprising, given that its oxyanion-stabilizing role occurs through backbone atoms; however, as discussed below, its carboxyl side chain also appears to have a catalytic role, particularly in the activated state.

**Conformational changes in the transition to the active state.** The overall structure of the enzyme does not change significantly after autocleavage (1.1 Å r.m.s.d.), but there are conformational changes at the junction between the α- and β-subunits. First, residues 141 and 142 at the C terminus of the newly created α-subunit become disordered. Second, the last turn of the C-terminal helix-α5 of the α-subunit (residues 135 to 140) bends "away" from the enzyme by roughly 60° (Fig. 3a). This conformational change repositions Tyr 137 relative to the active site strand-β1, from a hydrophobic environment in the proenzyme, to one where the bent conformation of helix-α5 is stabilized by hydrogen bonds between its side chain hydroxyl and the protein backbone, thus allowing access to the active site (Supplementary Fig. 7).

These changes in the cleaved state uncover a 13 Å deep, surface-accessible channel with Cys 143 sitting at its base (Fig. 3b). The channel is composed of a constellation of mainly hydrophobic residues, presumably to accommodate the ceramide acyl chains, while directing the scissile amide bond towards the active site (Fig. 3c). Additionally, an internal hydrophobic cavity lateral to Cys 143 is observed upon autocleavage (Fig. 3b).

Based on the geometry of the channel and the position of Cys 143 in the active structure of human-aCDase, we modeled how ceramide might sit in the active site. The coordinates for a ceramide molecule with optimal lipid chains lengths were initially

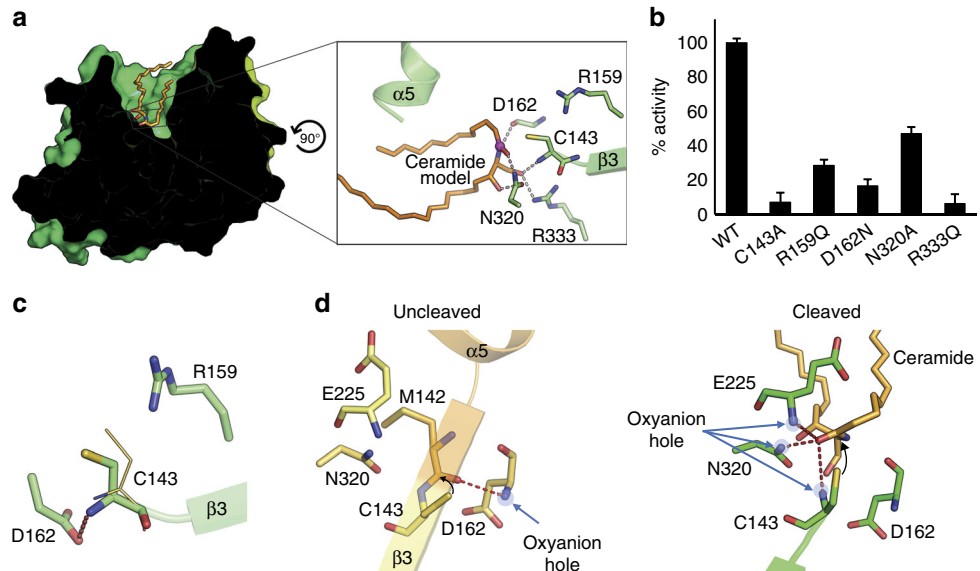

**Fig. 4** The active state of aCDase. **a** Docking of C12 ceramide into the active site of aCDase. Left, surface representation of the active human enzyme with the docked ceramide shown as orange sticks. The box on the surface is shown zoomed in the right panel. Right, shown are active site residues (green sticks) that make potential interactions (dashed pink lines) with the modeled ceramide. The carbonyl carbon in ceramide attacked by Cys 143 is highlighted as a purple sphere. Hydrogen bonds within the protein active site are omitted for clarity. **b** Ceramidase activity of the active site mutants of aCDase on ceramide-containing anionic liposomes at pH 4 in the presence of the co-factor saposin-D. One hundred percent activity for wild-type enzyme corresponds to 0.54 μM ceramide hydrolyzed per nM protein per hour. Data are the means and s.d.'s of eight replicates. **c** Shift in position of Cys 143 upon activation. The position of the modeled Cys 143 rotamer from the inactive state is shown as a thin yellow stick. Residues in the active structure are shown as green sticks. Asp 162, which was further away from the Cys 143 backbone amide in the inactive state, is closer in the active state to form a weak salt-bridge (red dashes, 3.4 Å). Conversely, Arg 159 no longer forms a hydrogen bond with Cys 143 in the active state. **d** Comparison of the direction of the nucleophilic attack by Cys 143 on the target amide during autocleavage versus substrate hydrolysis. The two views are in roughly the same orientation. The interaction of the scissile peptide carbonyl oxygen with potential oxanion hole residues in the respective states are indicated (dashed red lines)

manually docked into the active site by constraining the distance between Cys 143 and the scissile peptide. Several initial orientations were tested and energy-minimized, allowing the substrate to adjust its position, but keeping the protein fixed. The result for the best pose with regards to maximizing favorable interactions with the protein is shown in Fig. 4a. Ceramide sits in the active site channel with its lipid tails surrounded by hydrophobic residues. At the bottom of the channel, the ceramide head group hydrogen bonds with several residues including Arg 333, Asn 320, Asp 162, Glu 225, and the N-terminal amino group of Cys 143.

The shape of the substrate binding channel appears to be specific for ceramide, as other membrane-resident lipids with bulky head groups such as sphingomyelin, phospholipids, and cerebrosides, would result in steric clashes were they to bind in a manner similar to the modeled ceramide (Supplementary Fig. 8). However, certain lipids were reported to inhibit hydrolysis or synthesis of ceramide by aCDase;[40] therefore, the interaction of various lipids with the enzyme merits further investigation. Diacylglycerol (DAG), a lipid structurally related to ceramide, was also modeled in the active site, but it forms fewer hydrogen bonds with the protein compared with ceramide (Supplementary Fig. 9A, B). Therefore, the affinity for DAG is expected to be lower than ceramide. In support of this, aCDase showed a 12-fold lower hydrolysis rate for DAG compared with ceramide (Supplementary Fig. 9C).

**Catalytic mechanism of substrate hydrolysis.** Docking of ceramide allowed us to propose a mechanism for substrate cleavage, which appears to be distinct from autocleavage due to the conformational changes that occur at the active site following

autocleavage, and substrate geometry considerations. In addition to uncovering the substrate-binding site upon autocleavage, the concomitant conformational change to the α-β junction causes strand-β3 containing Cys 143 to shift (Fig. 4c). This moves Cys 143 away from Arg 159 breaking their putative hydrogen bond, suggesting that for substrate cleavage Arg 159 may not act as the general base. Instead, the general base in the activated state is mostly likely to be the newly formed N terminus of the β-subunit, as has been found in other Ntn-hydrolases, such as Acyl coenzyme A:isopenicillin N acyltransferase[30, 41]. Repositioning of Cys 143 also results in the Asp 162 carboxyl being better positioned to stabilize the positive charge of the N terminus and, accordingly, its calculated pKa has shifted to about 3 in the activated state. The tightly bound water, W1, from the proenzyme is no longer present, as it would clash with Cys 143 and presumably is no longer required because the active site is now accessible to the outside environment.

The main variable in docking ceramide was the orientation of the carbonyl of the scissile peptide since the carbonyl oxygen forms a transient oxanion that must be stabilized during the reaction mechanism. Because of the orientation of the substrate-binding site relative to the position of Cys 143, the direction of nucleophilic attack on substrate is different from that of autocleavage. In autocleavage, nucleophilic attack occurs from 'above' the peptide bond, whereas during substrate cleavage attack must occur laterally (Fig. 4d). Consequently, the position of the scissile peptide carbonyl is different from that in autocleavage, as are the residues that stabilize the oxanion. In the modeled orientation of ceramide, the oxanion hole is provided by the side chain nitrogen of Asn 320, the backbone nitrogen of Glu 225 and possibly the free N terminus of Cys 143, whereas the Asp 162 backbone nitrogen has this role during

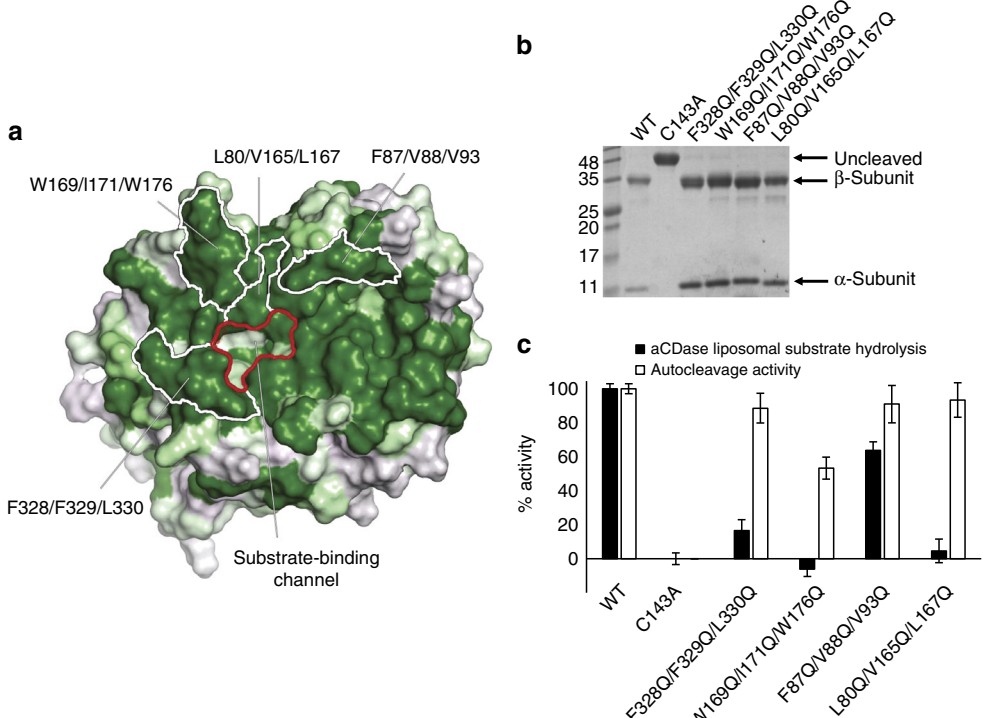

**Fig. 5** The hydrophobic surface of aCDase. **a** Surface view of cleaved aCDase with the hydrophobic regions colored green. Mutated residue clusters are lassoed in white, while the substrate binding site opening is outlined in red. **b** Reducing SDS-PAGE of the purified aCDase surface mutants showing that all mutants could autocleave. **c** Activity measurements of aCDase surface mutants on ceramide-containing anionic liposomes at pH 4 in the presence of the activator protein saposin-D compared with their autocleavage rates. For the liposomal assay, 100% activity for wild-type enzyme corresponds to 0.54 μM ceramide hydrolyzed per nM protein per hour. For autocleavage, 100% activity correspond to $k = 0.1673$ (fraction cleaved per hour). Data for the ceramidase activity assay are the means and s.d.'s of eight replicates

autocleavage. Based on these considerations, Supplementary Fig. 10 depicts a potential mechanism of substrate hydrolysis.

To assess the validity of the ceramide modeling, we made several active site mutations (Fig. 4b). R333Q impaired substrate hydrolysis almost as much as C143A, highlighting its importance in both orienting Cys 143 and potential interaction with the substrate; the same mutation had a weaker effect on autocleavage (Fig. 2c). Likewise, D162N had a strong effect due to its potential role in stabilizing the N-terminal positive charge. R159Q also had a significant impact on activity—although Arg 159 does not hydrogen bond with Cys 143 in the active state, it is still likely to be important in positioning the nucleophile through Van der Waal's interactions. Finally, N320A had the weakest effect, presumably because its posited role as the oxyanion hole could also be performed by Glu 225 and/or the N terminus (Fig. 4d).

**A hydrophobic surface in aCDase regulates ceramide intake**. In addition to the hydrophobic residues that line the substrate-binding channel, the surrounding region is also strikingly hydrophobic. This surface is comprised of 39 residues from both the α- and β-subunits, which together encompass ~ 2914 Å$^2$ of exposed hydrophobic area (Fig. 5a). The proximity of this surface to the substrate-binding site, its evolutionary conservation (Supplementary Fig. 4), its exposed hydrophobicity, and its shallow bowl-like shape suggests it has an important function.

To assess the importance of this region, we mutated residues on it and compared the activity of the mutants versus wild-type protein on ceramide-containing anionic liposomes at pH 4 to mimic the native conditions of the lysosome. Because of its size, we mutated groups of three spatially clustered residues (Fig. 5a), reasoning that single mutations may be ineffective. Although aCDase is active on

its own, optimal catalysis is achieved in the presence of saposin-D[31], which was purified and included in the assay. To control for mutations that perturb the protein fold or stability, we also measured their autocleavage rate. All surface mutants underwent proper proteolytic autocleavage in vitro (Fig. 5b), suggesting that they were properly folded and active. All mutants showed reduced ceramide hydrolysis compared with wild-type in the liposomal assay (Fig. 5c). In particular, hydrophobic patches mutated near the substrate binding channel (F328Q/F329Q/L330Q and L80Q/V165Q/L167Q) had strong effects, whereas mutation of a site further from the substrate-binding site (F87Q/V88Q/V93Q) reduced the substrate hydrolysis rate by 50%. This indicates that the hydrophobic surface outside of the substrate-binding site has a role in possibly membrane attachment, accessing substrate in a lipid environment, or saposin-D binding. The addition of saposin-D to wild-type aCDase in the liposomal assay significantly increases its activity (Supplementary Fig. 9D), confirming its role in enhancing aCDase activity[42].

Interestingly, mutation of the L4-6 loop in the β-subunit (W169Q/I171Q/W176Q) that was observed in very different conformations in the two inactive structures (Supplementary Fig. 1B, C) not only significantly reduced ceramide hydrolysis, but also reduced autocleavage activity (Fig. 5c and Supplementary Fig. 5). Thus, L4-6 is important for protein stability and could be an important component of activation. Importantly, L4-6 harbors Trp 169, a residue mutated in FD (see section below), further supporting its functional significance.

**Disease mutations**. Mutations in aCDase are responsible for mainly two recessive genetic disorders: FD and SMA-PME[11, 12]. To date, 24 different missense mutations in aCDase have been

 

associated with the incidence of the two diseases[43]. The active structure of human-aCDase allowed us to map these mutations to rationalize their potential effects. Figure 6 and Table 2 show the location of 16 different mutations that cause FD and 2 that cause SMA-PME.

Thirteen mutations localize to the protein interior, likely disrupting its fold or stability. Two mutation positions, N320 and R333, are active site residues whose mutation likely inhibit autocleavage and/or substrate hydrolysis. N320 stabilizes the N terminus of Cys 143 through hydrogen bonding with its side chain oxygen, whereas its side chain nitrogen provides the oxyanion hole for substrate hydrolysis and stabilizes the position of Glu 225, also important for oxyanion hole formation. N320 is found mutated to either aspartate or serine, which would disrupt the functional requirement for an asparagine side chain at this position. R333 hydrogen bonds to N320 and based on the substrate modeling above, is predicted to be important for

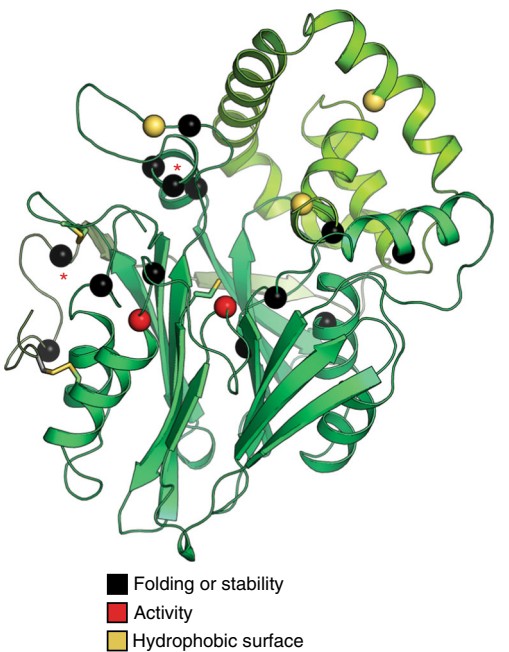

**Fig. 6** Structural mapping of disease mutations. Disease mutations are colored based on their expected effect on the enzyme. Black spheres represent disease mutations that likely disrupt the folding or stability of the enzyme, red spheres are mutations that affect the active site of aCDase, and yellow spheres are mutations that affect the hydrophobic surface. The α-subunit of the mature enzyme is in light green and the β-subunit is in green. Cys 143 is shown as sticks. Locations of observed SMA-PME missense mutations in the structure are labeled with a red asterisk; all other locations are missense mutations causing FD

engaging ceramide. Its mutation to the shorter side chains of histidine or glycine would hinder its function.

V97, F136, and W169 are all mutation positions found on the expansive hydrophobic surface identified above. V97D could inhibit the interaction of aCDase with negatively charged liposomes, whereas V97G likely destabilizes helix-α2 in the α-subunit in which it resides. As mentioned, W169 resides in L4-6 and probably affects the stability of the enzyme and possibly its transition to the activated state, as well as disrupting potential membrane or substrate interactions. F136 is located near the lipid tails of the modeled substrate and at the α-β interface where the F136L mutation could destabilize the heterodimer and substrate interactions.

Finally, we analyzed the potential effects of additional rare variants that are found in aCDase, but not necessarily connected to a disease phenotype (Supplementary Table 1). Although most such variants probably have a benign effect on the overall fold, stability, and activity of the enzyme, we predict that at least 27 other variants will have a harmful impact leading to malfunctioning ceramidase activity.

## Discussion

The structures of mammalian aCDases determined here provide the first insight into the structural basis for the lysosomal breakdown of the critical sphingolipid, ceramide, which is the only source for cellular sphingosine, the base moiety for all mammalian sphingolipids. The αββα-fold of the aCDase β-subunit definitively categorizes it as a member of the Ntn-hydrolases. The variable α-subunit of Ntn-hydrolases is present in aCDase as a disulfide bonded N-terminal linker followed by a globular five-helical subdomain bound to the β-subunit through a network of hydrophobic interactions. The only other known structure of a mammalian ceramidase is of human nCDase[44]. However, nCDase is a single-pass transmembrane protein expressed mainly in epithelial cells of the intestine and colon, and thus acts at a higher pH and on a different membrane environment than aCDase[6, 45, 46]. It belongs to a different structural family and catalyzes ceramide hydrolysis with a distinct mechanism that depends on active site zinc ions[44].

Activation of aCDase requires autocleavage by Cys 143. Our structures confirm that this reaction must occur intramolecularly and suggests deprotonation of Cys 143 can occur either by Arg 159 or a bound water molecule. Deprotonation of the active site Cys (or Ser, Thr) by a nearby water molecule was also proposed for other Ntn hydrolases. In the structure of the cephalosporin acylase (CA) precursor[47], a water molecule is present at the same position as in the aCDase proenzyme and similarly hydrogen-bonded, with a backbone carbonyl taking the place of the Thr 141 side chain (Supplementary Fig. 11). As with our mutation of Thr 141, the importance of this water molecule was confirmed in CA by a mutation that distorts the active site and inhibits autocleavage[48]. A conserved water molecule was also proposed to act

| Table 2 Predicted effects of aCDase mutations found in FB and SMA-PME | | | | | | | | | |
|---|---|---|---|---|---|---|---|---|---|
| *Mutations predicted to affect the folding or stability of the protein* | | | | | | | | | |
| Y36C | T42A | T42M | V97E | V97G | E138V | G168W | T179I | E180K | L182V |
| T222K | R226P | G235R | G235D | R254G | D331N | P362R | P362T | | |
| *Mutations affecting the hydrophobic surface of the protein* | | | | | | | | | |
| | | | F136L | W169R | | | | | |
| *Mutations inhibiting the activation of the enzyme* | | | | | | | | | |
| | | N320D | N320S | R333G | R333H | | | | |

 

as a general base in the autocleavage of proteasome β- subunits[49]. However, this mechanism does not extend to all Ntn hydrolases; for instance, an aspartate acts as the base in glycosylasparaginase autocleavage[50].

ACDase autocleavage results in conformational changes that uncover a narrow hydrophobic channel leading to the active site that likely accommodates ceramide substrates. This mode of active site exposure was observed previously in nCDase[44] and the Acyl coenzyme A:isopenicillin N acyltransferase from *P. chrysogenum*[41]. However, in nCDase, a larger 20 Å tunnel is uncovered which could explain the preference of this enzyme towards ceramide substrates of different acyl chain lengths (C12-ceramides for aCDase versus C16-ceramides for nCDase) (Supplementary Fig. 12)[28, 51]. A hydrophobic cavity lateral to Cys 143 is also observed upon autocleavage (Fig. 3b). However, its relevance is unclear as the location of the catalytic cysteine and oxyanion hole impose a constraint on ceramide orientation, which precludes interaction with this lateral cavity. Neither can the bulkier head groups of lipids such as a glucosylceramide or sphingomyelin interact with this cavity, as they would be located on the opposite side (Supplementary Fig. 8).

Although enzymatic catalysis employs Cys 143 for both autocleavage and substrate hydrolysis, the precise mechanism is different with regards to the direction of nucleophilic attack because of the differences in position of the scissile peptide in the proenzyme versus the incoming substrate. A consequence of this is that the oxyanion hole is provided by distinct residues depending on whether autocleavage or substrate hydrolysis takes place. An analogous switching of the oxyanion hole was observed previously in the fungal enzyme, Acyl coenzyme A:isopenicillin Nacyltransferase[41]. In addition, slight changes to the aCDase active site upon autocleavage suggested a difference in the general base of the reaction mechanism from an arginine or water molecule in the proenzyme to the newly formed β-subunit N terminus in the activated protein. Further experiments will be required to verify these notions.

The overall positive charge of different acidic hydrolases, such as acid sphingomyelinase (ASMase), is important for binding to intra-lysosomal anionic vesicles where lipid degradation occurs[52]. ACDase was previously shown to preferentially bind to anionic lipid vesicles compared with neutral ones, and Elojeimy et al.[53] demonstrated that cationic amphiphilic drugs disrupt this electrostatic interaction in the lysosome leading to dissociation and proteolytic degradation of the enzyme[42]. It is thus likely that aCDase uses its surface-exposed basic residues to interact with lysosomal membranes. Although there are a total of 34 positively charged residues (10 arginines and 24 lysines), we could not pinpoint specific charged patches on the protein responsible for binding to anionic liposomes, and simple electrostatic association of the enzyme to a lipid membrane may not be sufficient to access embedded lipid substrates. Thus, we propose a role for the large hydrophobic surface surrounding the aCDase active site in accessing membrane lipids. The presence of a hydrophobic surface on aCDase was previously suggested by Al et al.[54], where they observed that aCDase dimers can be dissociated with addition of Triton X-100. In addition, Linke et al.[42] suggested a mechanism by which aCDase binds to anionic lipids and uses hydrophobic patches to partially penetrate the hydrophobic layer of membranes to access substrates.

The requirement of saposin-D for full aCDase activity was shown to be important at lysosomal pH, as the hydrophobicity of this cofactor increases with decreasing pH[31, 55]. Some saposins have been shown to bind to and interact with their respective enzymes to enable full activity. For example, the interaction between saposin-C and glucocerebrosidase is essential for the association of the enzyme with liposomes and its lipase activity[34].

The structure of saposin-A with its associated hydrolase, β-galactocerebrosidase, was recently determined by Hill et al.[56]. Likewise, the ASMase saposin domain interacts with its catalytic domain and this contact is crucial for ASMase activity[57–59]. However, our biochemical and mutational analysis suggests that saposin-D does not form a complex with aCDase. Further experiments will be required to assess the precise mechanismby which saposin-D enhances aCDase activity.

ACDase has been shown to have causal roles in genetic diseases that inactivate the protein and the structure allowed us to predict the molecular basis for inactivation. Most disease mutations affect protein stability or folding, whereas a smaller subset affect enzymatic function. Importantly, the structures also help us evaluate the potential effect of numerous variants of aCDase discovered through sequencing projects. In cases where the enzyme becomes inactivated, recombinant aCDase is a potential treatment of maladies such as FD and SMA-PME, and the structure could provide an avenue of improvement for enzyme replacement therapy[24]. Finally, overexpression of aCDase has been linked to several cancers and there are ongoing attempts to design inhibitors that sensitize cancer cells to therapeutic treatments[4, 22]. In this regard, the detailed knowledge of the substrate-binding site and mechanisms of autocleavage and ceramide hydrolysis will pave the way for the development of therapeutic inhibitors.

## Methods

**Protein expression and purification.** Recombinant full-length aCDase was expressed as a secreted protein in *Sf*9 insect cells (Invitrogen) infected with baculovirus. The endogenous signal peptide comprising the first 21 residues was replaced by the melittin signal peptide (MKFLVNVALVFMVVYISYIYA) followed by a hexahistidine tag (DRHHHHHHKL). Constructs encompassed residues 22 to 395 of homologs from human (UniProt Q13510 variant Ile93Val), nmr (*Heterocephalus glaber*, UniProt A0A0P6JG37 variant Asn348Ser), or cmw (*Balaenoptera acutorostrata scammoni*, RefSeq XP_007174053). The latter two homologs were codon-optimized. All constructs and mutants were verified by sequencing. aCDase was isolated from culture media using nickel-nitrilotriacetic acid (Ni-NTA) resin (Thermo Fisher Scientific) and further purified by size exclusion chromatography on a Superdex 200 column (GE Healthcare) in 15 mM Tris-HCl pH 7.5 with 100 mM NaCl. Proteins for crystallization were either inactive mutants (Cys143Ala) or wild-type versions, which first underwent proteolytic autocleavage at 37 °C for 24–60 h. Sodium acetate (100 mM) pH 5.0 was added for autocleavage; proteins were subsequently exchanged back into the neutral buffer and concentrated to 10 mg mL$^{-1}$.

Recombinant human saposin-D (prosaposin residues 405–486, UniProt P07602) was expressed in *Escherichia coli* strain Rosetta-gami B (Novagen) with a hexahistidine tag (MSYYHHHHHHDYDIPTTENLYFQGAMGS) and purified on Ni-NTA resin. The tag was cleaved off with TEV protease, leaving a GAMGS overhang at the N terminus. Saposin-D was further purified on a Superdex 75 column (GE Healthcare) in 20 mM sodium acetate buffer pH 5.0 with 100 mM NaCl and concentrated to 7.5 mg mL$^{-1}$.

**Crystallization and data collection.** Crystals were grown by sitting or hanging drop vapor diffusion at 22 °C. Inactive nmr aCDase crystallized in 100 mM MES pH 6.0, 10% glycerol, 5% PEG 1000 and 30 % PEG 600. Inactive cmw aCDase crystals were obtained in 100 mM citric acid pH 3.5 and 3 M NaCl. Autocleaved human-aCDase was incubated with 1 mM ceranib-2 (Sigma) and 1 mM Triton X-100, and crystallized in 100 mM sodium phosphate-citrate pH 4.3, 200 mM Li$_2$SO$_4$, and 20% PEG 1000. Crystals were soaked in well-solution supplemented with 20% glycerol and heavy atom compounds: 500 mM NaI for nmr protein crystals (20 s) or 500 mM lithium 5-amino-2,4,6-triiodoisophthalate for the cmw homolog (20 s). X-ray diffraction data were collected at 100 K on beamline 08ID-1 with a Rayonix MX300 CCD detector or on beamline 08B1-1 with a Rayonix MX300HE CCD detector at the Canadian Macromolecular Crystallography Facility, Canadian Light Source. Data were processed by HKL2000[60].

**Structure determination and refinement.** The structure of nmr aCDase was solved by iodine single-wavelength anomalous diffraction using Autosol[61] in Phenix[62], manually built in Coot[63] and refined against a dataset from a non-soaked crystal. The structures of human and cmw enzymes were obtained by molecular replacement using Phaser[64] in Phenix. Refinement was carried out by phenix. refine[65] with the following parameters: anisotropic non-hydrogen B-factors for nmr aCDase, non-crystallographic symmetry restraints for its two homologs, translation-libration-screw for the human-aCDase. Crystallographic data collection and refinement statistics are presented in Table 1. Structural images were prepared with PyMOL (The PyMOL Molecular Graphics System, Version 1.3 Schrödinger,

LLC). No density for ceranib-2 was observed in the electron density of human-aCDase. Calculation of protein side chain pKa values was carried out using the PDB2PQR server[66].

**Ceramide docking**. The coordinates for various initial orientations of a C12 ceramide molecule were manually docked into the active site of the human-aCDase structure in Coot[63]. The distance between the Cys 143 sulfur atom and the ceramide carbonyl carbon was fixed at 3 Å. Residues near the active site were fixed to improve the stability of the energy minimization procedure using Groningen Machine for Chemicals Simulations (GROMACS) 4.6.2 package[67] with GROMOS 96 force field[68]. The aCDase topology file was generated by using the GROMOS 53A6 force field. The ceramide topology file was generated with the same force-field as the protein by the Automated Topology Builder Server[69]. Each aCDase-ceramide complex was placed and solvated in the center of a dodecahedron box with an adjusted 10 Å distance from each side of the box. The system's charge was neutralized by the addition of a single chloride ion. The energy minimization for each system (nsteps = 50 000) was done by the steepest descent algorithm. The Particle Mesh Ewald method was used to calculate electrostatic interactions[70]. The cut-off distances for the Van der Waals interactions, the Coulomb interactions, and the short-range neighbor list were all set to 10 Å.

**Liposome-based enzymatic assay**. The liposome-based enzymatic assay was adapted from the protocol described by Alayoubi et al.[23]. Liposomes were prepared by drying the lipid mixture, resuspending it in water, multiple freeze–thawing cycles, followed by extrusion through 100 nm polycarbonate filters. Negatively charged liposomes consisted of 25% mole Bis(monoacylglycero)phosphate, 45% 1,2-dimyristoyl-sn-glycero-3-phosphocholine, 20% cholesterol, and 10% 12:0 ceramide. aCDase at a concentration of 125 nM was mixed with 1.25 mM of anionic liposomes, for a total ceramide concentration of 125 μM, in assay buffer (100 mM NaCl, 20 mM sodium acetate, pH 4). The reaction was incubated at 37 °C for 1 h, in the presence of 1.25 μM saposin-D, before incubating the mixture at 95 °C for 5 min to stop the reaction and adding 20 mM Triton X-100 for solibilizing the liposomes. The amount of ceramide hydrolyzed was determining by measuring the amount of free fatty acid released using Cayman Chemical Company's Free Fatty Acid Fluorometric Assay Kit (Item Number 700310). The contribution of the different components was tested and presented in Supplementary Fig. 13.

Hydrolysis of DAG (dipalmitoylglycerol) and ceramide was compared in a simplified Triton X-100-based setting: 500 μM of substrate were incubated with 250 nM of aCDase, 50 μM of saposin-D, and 10 mM of Triton X-100 for 1 h at 37 °C, before stopping the reaction at 95 °C for 5 min and measuring the amount of free fatty acid released.

**Autocleavage assay**. aCDase at a concentration of 1 mg mL$^{-1}$ was incubated in cleavage buffer (100 mM NaCl, 20 mM sodium acetate, pH 5) at 37 °C. One microgram of sample was taken every 4 h for 48 h. Samples were analyzed by SDS-polyacrylamide gel electrophoresis and the intensities of the bands of uncleaved and cleaved aCDase were quantified by the ImageJ software[71, 72]. Autocleavage rates were determined using a one-phase decay nonlinear regression fit.

**Data availability**. Data supporting the findings of this manuscript are available from the corresponding author upon reasonable request. Coordinates and structure factors were deposited in the Protein Data Bank under the accession codes 5U81 (nmr-aCDase), 5U84 (cmw-aCDase), and 5U7Z (human-aCDase).

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

## Acknowledgements

We thank the staff at the Canadian Macromolecular Crystallography Facility, Canadian Light Source, which is supported by the Natural Sciences and Engineering Research Council of Canada, the National Research Council Canada, the Canadian Institutes of Health Research, the Province of Saskatchewan, Western Economic Diversification Canada, and the University of Saskatchewan. We thank Dr. John Silvius for assistance in preparation of liposomes. B.N. was supported by an operating grant from the Canadian Institutes of Health Research (CIHR grant MOP-133535). A. Gorelik was supported by the CIHR Strategic Training Initiative in Chemical Biology and a CREATE Training Program in Bionanomachines (NSERC). A. Gebai was supported by the CIHR Strategic Training Initiative in Chemical Biology.

## Author contributions

A.Gebai, A.Gorelik, K.I., and B.N. designed the project. A.Gebai, A.Gorelik, and B.N. wrote the manuscript. A.Gebai and A.Gorelik performed all the structural and enzymatic experiments, and contributed equally to this work. A.Gebai and Z.L. performed the docking experiments. K.I. performed all cell culture experiments.

## Additional information

**Competing interests:** The authors declare no competing interests.

