## [Peer Review File · Nature Communications]

Reviewers' comments:

Reviewer #1 (Remarks to the Author):

The paper by Gebai et al, describes the structure of acid ceramidase, an important enzyme in sphingolipid metabolism that when defective results in the severe diseases Farber disease and SMA-PME. The description of the structure is fine although the quality of the data and refinement cannot be judged until the Table 1 and validation reports are provided. Furthermore, although the structure itself is informative the description of the mechanisms of self-cleavage and substrate processing need improvement. I also have significant concerns regarding the activity assays and the description of SapD relevance.

Major comments

Self-cleavage and catalytic mechanism

As the authors state on p9 the mechanism of both acid ceramidase self cleavage and substrate hydrolysis has been previously proposed based on homology modelling. Although, the structure provided here supports this model there are several concerns regarding the details of the proposed mechanism and substrate binding.

1. The authors state in several places what the calculated pKa of several active site residues are but do not describe in their methods how these were calculated. Residues at an active site can have extremely different pKa values from those that would normally be predicted. The authors need to describe what algorithms they used for these calculations.
2. Why wasn't mutation of T141 carried out as part of Figure 3C? The discussion on p10 of the importance of this residue suggests this would be a critical mutation to test.
3. The discussion of Residue D162 on page 11 is very confusing. The sidechain seems to be important for cleavage but the sidechain doesn't seem to be interacting with critical residues. Then the potential role in pH dependence is mentioned but not tested in this work. This section needs to be rewritten to clarify what the authors think the role of D162 is or else remove this section.
4. The authors speculate regarding the position of the Cys143 sidechain required for self-cleavage and need to model the sidechain in the uncleaved form as it was mutated to alanine in the uncleaved structure. This would be acceptable except that then the authors make a detailed comparison of the position of the C143 sidechain in the cleaved form with what is only a modelled sidechain position in their uncleaved structure including description of a 40° rotation and an illustration in Fig 5C. The description of these conformational changes doesn't seem to provide much greater insight than comparison with previously determined structures of other Ntn-hydrolases.
5. The description of the enzyme mechanism on p14 is difficult to follow without a figure. It would be useful to include a schematic of what the authors think the catalytic mechanism is and how the relevant active site residues contribute to the cleavage of the substrate. Clear reference to other papers describing similar mechanisms would also be useful such as those used in describing structure 2X1C.
6. Can the authors determine from their structure why this enzyme is optimal at acidic pH? The authors could use their activity assay and their catalytic mutants to determine if the pH profile of the activity has shifted.
7. In the methods section it is stated that the cleaved human AC was crystallised in the presence of ceranib-2 (an inhibitor of AC). Was there any evidence in the electron density of this bound in the active site?
8. In the methods the enzymatic assay describes the inclusion of anionic liposomes but doesn't describe how these liposomes were made. These methods should be included.
9. In the methods it is stated that 20mM Triton X-100 was added "for solubilising the liposomes". Why did these need solubilising? Doesn't this mean the ceramide substrate would be accessible to AC without any requirement for SapD to extract it from the liposome?
10. As the method used to monitor ceramide hydrolysis was to measure the amount of free fatty

acid present, then controls need to be included showing how much fatty acid is detected in the absence of AC.

Ceramide docking

11. What are the B-factors of the residues surrounding the docked ceramide substrate? Is it sensible to have kept these fixed while docking substrate? The authors state (p12) that the ceramide sits "snugly" in the active site. Does it fit including full van der Waals radii or is it likely that there would be some sidechain rearrangement to accommodate ceramide?

12. Figure 5A is not very informative. A cut-through like shown in Fig 4B would be more useful.

13. On p13 the authors conclude that the channel is specific for ceramide and other lipids containing bulky headgroups wouldn't fit. But looking at the cut-through in Fig 4B there is a big lateral cavity identified that looks like it could accommodate exactly such a headgroup. Did the authors try docking an alternative lipid? Did the authors try their activity assays with other lipids? These would both be important additional information to support the claim that the pocket is highly specific.

14. Does the structure explain why AC can't process diacylglycerol as this would surely fit the active site?

Hydrophobic surface

15. The authors speculate on the importance of the hydrophobic surface near the active site and state that it may be critical for interaction with bilayers/liposomes. However, the liposomes and membranes that it is interacting with are charged – indeed the liposomes used in their activity assay are anionic. The authors need to clarify why they think that a highly hydrophobic surface would be optimal for interacting with an anionic lipid layer.

16. In the discussion (p19) the authors suggest that the hydrophobic area suggests more invasive binding of aCDase to lipid vesicles. Can the authors please explain and justify this statement? Similarly the last part of this paragraph seems to be confused about whether "nonspecific binding" to membranes occurs or whether there are important charged patches.

Sapoin D dependency

The manuscript makes several references to the importance of SapD for AC processing of ceramide. However, there is no data in this paper to support that and no insight into how this may function. The authors quote previous work saying SapD enhances AC activity but have not discussed other work by Linke et al JBC, 2001 identifying that not only SapD but also SapC and SapA can stimulate ceramide hydrolysis by AC. The authors included SapD in their activity assays but did not show what the activity was without SapD, an unacceptable omission.

17. If the discussion regarding saposins is to be retained then additional experiments have to be included showing a comparison of AC activity in the presence and absence of SapD to validate that the AC they have produced needs SapD to process ceramide in their assays. Furthermore they should compare the activity in the presence of other saposins such as SapA or SapC to support arguments relating to specificity.

18. The text at the end of the Introduction is far too speculative where the authors state that SapD "appears to act independently by facilitating ceramide availability" as they have provided no data in this manuscript to test this let alone data that support this statement.

19. The discussion on p16 regarding the role of SapD is overstated based on the data presented here. The authors state they cannot identify any interaction between AC and SapD and do not show enhanced activity themselves but then state that SapD must be disrupting the lipid membrane. Although this may be correct this work has not contributed any understanding to this process and so should be removed unless the authors can show both enhanced activity with SapD and association of SapD with their liposomes.

20. The discussion on p20 regarding saposin function is not clear and is poorly described. If this discussion is to remain in the paper then there needs to be a clearer explanation of why it's

important. Are they describing the difference between solubilising vs liftase saposin models or are they discussing open vs closed saposins. And as stated above, not detecting an interaction between SapD and AC does not alone provide evidence to support that SapD disrupts membranes.

The L4-6 loop

21. Results, p7: The authors mention that the L4-6 loop conformation is different and even state that this difference is interesting but then state it's involved in a crystal contacts meaning interpretation of its importance is difficult. The subsequent discussion (p15) about how this loop might be interesting also seems to be wrong as the authors state that mutation of this loop (W169Q/I171Q/W176Q) reduced autocleavage activity but Fig 6B seems to show cleavage is perfectly normal and so this loop can't play a role in protein stability. Unless the authors can argue otherwise, the entire discussion of this loop should be removed from the results and discussion.

Disease-relevant mutations

22. Can the authors use their structure to understand or speculate as to why the mutations they have mapped would lead to two different diseases: Farber disease versus SMA-PME. The SMA-PME mutations shown in Fig 7 are identified as causing misfolding, the same as for several Farber disease mutations. What would be the molecular mechanism by which misfolded AC would lead to these different disease outcomes?

23. The last paragraph of the discussion of these disease mutants is very weak. It is unclear how the authors can conclude that these mutations would lead to a "malfunctioning ceramidase" while not being connected to a disease phenotype and are described here as having only a benign effect on fold stability or activity. This needs to be rewritten or removed.

Minor points

24. The title should be changed from the "Molecular Mechanism" of Acid Ceramidase to "The structure" of Acid Ceramidase. The work relating to the mechanism is not as novel or convincing as the structural work and so should not be the focus of the title.

25. Results, p6: The % sequence identity of naked mole rat and common minke whale to human acid ceramidase should be included.

26. Results, p7: The sequence identity between nmr and cmw aCDase is quoted to describe the similarity of the overall structures. The RMSD between these structures should also be included here.

27. Panel C of Figure 1 is unnecessary: the differences between these two structures is very minimal and doesn't seem to have any importance biologically so this panel should be removed.

28. Figure 2 is not very informative and should be removed. The discussion of the comparison with these other homologous proteins is not very informative either. The discussion of the differences in the alpha subunit don't lead anywhere – does lacking an alpha subunit completely change the access to the active site for example?

29. Results, p9: I think the authors mean "preceded by" not "followed by" when discussing the tight turn near residues 142 and the helix which forms the C-terminus of the alpha subunit.

30. What is the point of the inset panel in figure 3A showing a surface representation of the helix near C143?

31. Supp Figure 3 is difficult to understand and it's not clear from the text (p12) what the relevance is of this conformational change. If there is some importance to this movement it needs to be more clearly explained and the figure improved to make it clear how panels A and B relate to the overall structures.

32. Discussion, p18: Can the authors speculate as to why the neutral ceramidase would use an entirely different fold and mechanism to cleave the same substrate?

33. Discussion, p18: The authors state that neutral CDase possesses a larger tunnel that might accommodate different chain lengths. A Supp figure to show this would be helpful.

34. How is the highly hydrophobic surface of AC buried in the crystal packing? Is there dimer

formation?

35. Discussion, p20: The structure doesn't "explain" the molecular basis of protein inactivation in disease, it provides predictive power that can be tested in functional assays.

36. The two panels in Fig 7 are quite redundant. Panel B with labels identifying the two SMA-PME mutations would be just as useful.

Reviewer #2 (Remarks to the Author):

The present manuscript entitled "Molecular mechanisms of acid ceramidase" by Gebai and colleagues presents the first crystal structures of mammalian acid ceramidases in the proenzyme inactive and the cleaved active forms. The authors provide insights in the mechanism of activation and molecular function of the enzyme. The findings are novel and convincing. They are relevant to the field, as they will aid the design of acid ceramidase inhibitors.

Several issues remain:

1. Methods section is incomplete: No details are given on the presented Western Blots in the methods section or figure legends (Figs. 3 and 6). Importantly, it is not stated which antibody was used and against which part of the protein it was directed. It is also not expressively stated which species is shown (Fig. 3C: human, naked mole rat, common minkle whale?). Details regarding the expected molecular weight vs. the actual size should be discussed as well. In Figure 5B appropriate statistics are missing. An increase of the n number (currently n=3) is required. The same applies to Figure 6C.
2. The liposomal acid ceramidase assay employed in this study seems to be based on the assay described by Alayoubi et al., although this is not cited. The "source" of the protocol should be mentioned. Unlike previously reported acid ceramidase assays, the authors employ a fatty acid fluorimetric assay kit to quantify ceramide hydrolysis. Please validate this assay kit.
3. Acid ceramidase also catalyzes the reverse reaction (acetylation of sphingosine to ceramide) at slightly higher pH (Okino et al. 2003). The authors should discuss the identified structure also with regard to this mechanism.
4. The title seems vague and does not describe the content properly.
5. Farber Disease is abbreviated "FB" throughout the manuscript. The more common "FD" abbreviation would be preferable.
6. Page 7, line 7: The mentioned red arrow is in Figure 1B, not 1A.

Reviewer #3 (Remarks to the Author):

This is an excellent manuscript that describes the first crystal structures of human acid ceramidase and two homologs in activated and proenzyme forms, respectively. The structures provide important insight into Farber disease, are well refined, and may help drug discovery efforts given acid ceramidase is a proposed therapeutic target for cancer. Structural comparison of proenzyme and activated states reveals a conformational change upon activation to generate a new hydrophobic cavity to accommodate the membrane substrate ceramide. Modeling of ceramide into the activated state cavity provides a plausible catalytic mechanism that is supported biochemically. A different mechanism is proposed for autocleavage of the proenzyme that proposes an arginine residue or a bound water molecule as the catalytic base for the catalytic cysteine residue. Both mechanisms seem plausible. The only minor point is whether this type of mechanism is conserved in the Ntn-hydrolase family and if any other Ntn members conserve/utilize an Arg residue as a catalytic base, or the Thr or similar residue to stabilize a water molecule for hydrolysis.

Minor point:

1. How does the proposed autocatalytic mechanism for aCDase compare to other Ntn-hydrolase members with regards to the critical residues Arg159 and Thr141/water molecule?

Response to review points:

Reviewer 1:

1. **The authors state in several places what the calculated pKa of several active site residues are but do not describe in their methods how these were calculated. Residues at an active site can have extremely different pKa values from those that would normally be predicted. The authors need to describe what algorithms they used for these calculations.**

Calculation of pKa was carried out using the PDB2PQR server [66]. This is now added to the methods section.

2. **Why wasn't mutation of T141 carried out as part of Figure 3C? The discussion on p10 of the importance of this residue suggests this would be a critical mutation to test.**

In the revised manuscript, we introduce the T141V mutant into human aCDase and measured its effect on the autocleavage rate. Mutation of Thr 141 significantly reduced the autocleavage activity of aCDase, down to 40% of the wild-type's rate (Supp. Figure 5A), although the mutant was almost fully cleaved after a 65h incubation at acidic pH (Figure 2C). Since W1 potentially hydrogen bonds with 3 other atoms in the protein (the side chain of Asn 320, the backbone amine of Glu 225, and the side chain of Asn 320, and another water molecule), mutation of Thr 141 probably only partially dislodges W1 to inhibit autocleavage.

3. **The discussion of Residue D162 on page 11 is very confusing. The sidechain seems to be important for cleavage but the sidechain doesn't seem to be interacting with critical residues. Then the potential role in pH dependence is mentioned but not tested in this work. This section needs to be rewritten to clarify what the authors think the role of D162 is or else remove this section.**

We agree with the reviewer that this section was not clearly written and it has now been rewritten to show the important role of D162, particularly in the cleaved state. The side chain of D162 does indeed form important interactions with the new formed N-terminus at C143 in the cleaved enzyme due to the shift in position of C143 upon cleavage. This interaction was clearly depicted in Fig. 4C (formerly Fig. 5C). We suggest on pg. 13 that "Asp 162 is now better positioned to stabilize the positive charge of the amino-terminus and accordingly, its calculated pKa has shifted to about 3 in the activated state."

To further clarify the writing on pg. 11, we removed the discussion about D162's pKa and its role in pH dependence. The overall pH-dependence of the enzyme's activity was previously reported by other groups [29]. However, experimentally testing the pH dependence of D162 in this role is beyond the scope of this study.

- 4. The authors speculate regarding the position of the Cys143 sidechain required for self- cleavage and need to model the sidechain in the uncleaved form as it was mutated to alanine in the uncleaved structure. This would be acceptable except that then the authors make a detailed comparison of the position of the C143 sidechain in the cleaved form with what is only a modelled sidechain position in their uncleaved structure including description of a 40° rotation and an illustration in Fig 5C. The description of these conformational changes doesn't seem to provide much greater insight than comparison with previously determined structures of other Ntn-hydrolases.**

We modelled the C143 sidechain in the inactive structures to identify potential general bases for its deprotonation. We agree with the reviewer regarding the over analysis of the modelled C143 side chain and modified the writing as follows: “The side chain of Cys 143 is oriented similarly as the modelled conformation in the inactive state, but because of the conformation change to the α - β junction associated with autocleavage, strand β 3 containing Cys 143 has shifted.” The main point of this statement and of Figure 4C (formerly 5C) is to show that the backbone of Cys143 shifts upon cleavage and creates new interactions with D162.

Additionally, we removed the following speculative text regarding the Cys 143 rotamer modeling: “Thus, this rotamer is likely a good representation of the wild-type inactive state. In this position, the thiol sulfur atom is about 3 Å away from the carbonyl carbon of the preceding peptide bond, with a $\sim 130^\circ$ angle to the peptide plane. The ideal angle of nucleophilic attack on a planar carbonyl atom is typically 109° [40]. This slight deviation may explain why autocleavage is a relatively slow process.”

- 5. The description of the enzyme mechanism on p14 is difficult to follow without a figure. It would be useful to include a schematic of what the authors think the catalytic mechanism is and how the relevant active site residues contribute to the cleavage of the substrate. Clear reference to other papers describing similar mechanisms would also be useful such as those used in describing structure 2X1C.**

Figures for the mechanisms of autocleavage and substrate cleavage have now been added (Figures 2D and 5, respectively). The text on p.14 was modified to add a reference for other Ntn-hydrolase mechanisms.

- 6. Can the authors determine from their structure why this enzyme is optimal at acidic pH? The authors could use their activity assay and their catalytic mutants to determine if the pH profile of the activity has shifted.**

The catalytic mutations abolish the autocleavage activity of aCDase, making it difficult to study the pH-dependence of the enzyme.

7. **In the methods section it is stated that the cleaved human AC was crystallised in the presence of ceranib-2 (an inhibitor of AC). Was there any evidence in the electron density of this bound in the active site?**

We did not see any electron density for ceranib-2. This note has now been added to the methods section.

8. **In the methods the enzymatic assay describes the inclusion of anionic liposomes but doesn't describe how these liposomes were made. These methods should be included.**

The Methods section has now been updated with the liposome-construction protocol.

9. **In the methods it is stated that 20mM Triton X-100 was added "for solubilising the liposomes". Why did these need solubilising? Doesn't this mean the ceramide substrate would be accessible to AC without any requirement for SapD to extract it from the liposome?**

To solubilize free fatty acids and measure the amount released, we had to use detergent. This allowed the downstream enzymes in this coupled fatty acid detection assay to access the free fatty acids produced by acid ceramidase in the previous step, thus increasing the signal. Furthermore, this did not affect the reaction as the addition of detergent was done while the reaction was being incubated at 95°C to denature the enzyme and stop the reaction. This is now clarified in the Methods section.

10. **As the method used to monitor ceramide hydrolysis was to measure the amount of free fatty acid present, then controls need to be included showing how much fatty acid is detected in the absence of AC.**

See validation assay (**Reviewer Figure 1**).

11. **What are the B-factors of the residues surrounding the docked ceramide substrate? Is it sensible to have kept these fixed while docking substrate? The authors state (p12) that the ceramide sits "snugly" in the active site. Does it fit including full van der Waals radii or is it likely that there would be some sidechain rearrangement to accommodate ceramide?**

The B-factors values of the putative substrate binding residues are low relative to the surface of the protein (**Reviewer Figure 2**). The residues were fixed to eliminate artifacts from moving side chains causing clashes with the docked ceramide. The docking included van der Waals radii to eliminate atomic clashes, and an energy minimization step insured that all clashes were resolved.

12. **Figure 5A is not very informative. A cut-through like shown in Fig 4B would be more useful.**

We modified the left panel of Figure 4A (formerly 5A) to show a cut-through.

- 13. On p13 the authors conclude that the channel is specific for ceramide and other lipids containing bulky headgroups wouldn't fit. But looking at the cut-through in Fig 4B there is a big lateral cavity identified that looks like it could accommodate exactly such a headgroup. Did the authors try docking an alternative lipid? Did the authors try their activity assays with other lipids? These would both be important additional information to support the claim that the pocket is highly specific.**

There is indeed a lateral cavity near the active site. However, the location of the catalytic cysteine and oxyanion hole impose a constraint on substrate orientation. In the docked ceramide model, the acyl chain to be cleaved is facing towards that cavity, whereas the lipid head group is pointing in the opposite direction [shown in Supp. Fig. 8A]. There is a much smaller cavity on that side, but it is situated at the level of the ceramide hydroxyl group and is unable to accommodate bulkier head groups, such as a manually placed glucosylceramide [Supp. Fig. 8B] or sphingomyelin [Supp. Fig. 8C].

- 14. Does the structure explain why AC can't process diacylglycerol as this would surely fit the active site?**

Diacylglycerol (DAG) would fit into the active site, but it would form potentially only 7 hydrogen bonds with the protein, whereas ceramide establishes 11 (Supp. Fig. 6A, B). In the docked ceramide model, the head group hydroxyl contacts the side chains of Arg333 and Asp162 as well as the N-terminal amine of Cys143. The carbonyl oxygen interacts with the oxyanion hole comprising the side chain of Asn320, the backbone amide of Glu225 and the N-terminal amine of Cys143. These interactions would be maintained with DAG. However, the following hydrogen bonds are specific with ceramide: amide nitrogen with the backbone carbonyl oxygen of Asp162, and C3 hydroxyl with the side chains of Asn320 and Arg333. Therefore, the affinity for DAG is expected to be lower than that for ceramide. To verify this, we assayed hydrolysis of diacylglycerol and ceramide in a simplified Triton X-100-based setting, which showed a 12-fold reduction in activity on DAG compared to ceramide (Supp. Fig. 6C).

- 15. The authors speculate on the importance of the hydrophobic surface near the active site and state that it may be critical for interaction with bilayers/liposomes. However, the liposomes and membranes that it is interacting with are charged – indeed the liposomes used in their activity assay are anionic. The authors need to clarify why they think that a highly hydrophobic surface would be optimal for interacting with an anionic lipid layer.**

See response to point 16 below.

- 16. In the discussion (p19) the authors suggest that the hydrophobic area suggests more**

invasive binding of aCDase to lipid vesicles. Can the authors please explain and justify this statement? Similarly the last part of this paragraph seems to be confused about whether “nonspecific binding” to membranes occurs or whether there are important charged patches.

The discussion about the hydrophobic surface has now been modified:

The overall positive charge of different acidic hydrolases, such as acid sphingomyelinase, is important for binding to intra-lysosomal anionic vesicles where lipid degradation takes place [51]. ACDase was previously shown to bind better to anionic lipid vesicles as compared to neutral ones, and Eloheimy et al. (2006) demonstrated that cationic amphiphilic drugs disrupt this electrostatic interaction in the lysosome and lead to dissociation and proteolytic degradation of the enzyme [41, 52]. It is thus likely that aCDase uses its surface basic residues to interact with the lysosomal membrane. Although there is a total of 34 positively charged residues (10 arginines + 24 lysines), we could not pinpoint specific positively charged patches on the protein responsible for binding to anionic liposomes, and simple electrostatic association of the enzyme to a lipid membrane is not sufficient to access embedded lipid substrates. We propose a role for the hydrophobic surface surrounding the aCDase active site in this interaction. The presence of a hydrophobic surface on aCDase was previously suggested by Al et al. in 1989, when they observed that aCDase dimers can be dissociated with the addition of Triton X-100 [46]. Additionally, Linke et al. suggested a mechanism by which aCDase binds to anionic lipids and uses hydrophobic patches to partially penetrate the hydrophobic layer of the lipid membranes to access substrates [41]. Thus, the large hydrophobic surface is likely the feature allowing acid ceramidase to access lipids embedded in the membrane, by one or more of: embedding into the bilayer, disrupting it, or favoring lipid access to the active site.

- 17. If the discussion regarding saposins is to be retained then additional experiments have to be included showing a comparison of AC activity in the presence and absence of SapD to validate that the AC they have produced needs SapD to process ceramide in their assays. Furthermore they should compare the activity in the presence of other saposins such as SapA or SapC to support arguments relating to specificity.**

See Supp. Fig. 5B for necessity of SapD for optimal aCDase activity. The activity that other saposins have with aCDase is likely through similar mechanisms of actions shared among saposins. However, as we state in the text, SapA and SapC have specific enzymes they activate and form complexes with, and are thus unlikely to be the preferred activators of aCDase.

- 18. The text at the end of the Introduction is far too speculative where the authors state that SapD “appears to act independently by facilitating ceramide availability” as they have provided no data in this manuscript to test this let alone data that support**

this statement.

We removed from the introduction “which appears to act independently by facilitating ceramide availability” when discussing SapD and aCDase. We also removed speculation about SapD’s mechanism and modified the discussion to indicate only that there isn’t any data suggesting complex-formation with aCDase.

- 19. The discussion on p16 regarding the role of SapD is overstated based on the data presented here. The authors state they cannot identify any interaction between AC and SapD and do not show enhanced activity themselves but then state that SapD must be disrupting the lipid membrane. Although this may be correct this work has not contributed any understanding to this process and so should be removed unless the authors can show both enhanced activity with SapD and association of SapD with their liposomes.**

We now show enhanced activity of aCDase in the presence of SapD (See Supp. Fig. 5B) and combined with the inability to detect an interaction between the two proteins, we feel it is fair to suggest SapD possibly works by membrane disruption as has been found with other saposins. We clarify in the text that this is indeed speculation.

- 20. The discussion on p20 regarding saposin function is not clear and is poorly described. If this discussion is to remain in the paper then there needs to be a clearer explanation of why it’s important. Are they describing the difference between solubilising vs liftase saposin models or are they discussing open vs closed saposins. And as stated above, not detecting an interaction between SapD and AC does not alone provide evidence to support that SapD disrupts membranes.**

We removed the speculation part about SapD’s mechanism and removed the discussion on open vs closed conformations

- 21. Results, p7: The authors mention that the L4-6 loop conformation is different and even state that this difference is interesting but then state it’s involved in a crystal contacts meaning interpretation of its importance is difficult. The subsequent discussion (p15) about how this loop might be interesting also seems to be wrong as the authors state that mutation of this loop (W169Q/I171Q/W176Q) reduced autocleavage activity but Fig 6B seems to show cleavage is perfectly normal and so this loop can’t play a role in protein stability. Unless the authors can argue otherwise, the entire discussion of this loop should be removed from the results and discussion.**

Figure 6B showed that the loop (W169Q/I171Q/W176Q) mutation is fully cleaved after a 72h incubation. However, the autocleavage rate is significantly reduced to 53% of the wild-type’s (Figure 6C and Supp. Figure 5).

We have nonetheless removed the following speculative text “We speculate that the cmw aCDase L4-6 conformation where the loop covers and stabilizes the putative α - β junction represents a more downregulated state, while in nmr aCDase where this segment is essentially identical to that found in the active human structure, is an intermediate state of

activation.”

- 22. Can the authors use their structure to understand or speculate as to why the mutations they have mapped would lead to two different diseases: Farber disease versus SMA-PME. The SMA-PME mutations shown in Fig 7 are identified as causing misfolding, the same as for several Farber disease mutations. What would be the molecular mechanism by which misfolded AC would lead to these different disease outcomes?**

We unfortunately cannot provide a plausible explanation as to why certain mutations lead to a particular disease, since some of the disease-causing mutants are clustered together yet cause different phenotypes, and others are located on the surface of the protein and don't seem to have a significant effect.

- 23. The last paragraph of the discussion of these disease mutants is very weak. It is unclear how the authors can conclude that these mutations would lead to a “malfunctioning ceramidase” while not being connected to a disease phenotype and are described here as having only a benign effect on fold stability or activity. This needs to be rewritten or removed.**

This was an error in wording. We now clarify it by changing:

“Although most such variants probably have a benign effect on the overall fold, stability, and activity of the enzyme, we predict that at least 27 of them will have a harmful impact leading to a malfunctioning ceramidase.”

TO

“Although most such variants probably have a benign effect on the overall fold, stability, and activity of the enzyme, we predict that at least 27 other variants will have a harmful impact leading to a malfunctioning ceramidase.”

- 24. The title should be changed from the “Molecular Mechanism” of Acid Ceramidase to “The structure” of Acid Ceramidase. The work relating to the mechanism is not as novel or convincing as the structural work and so should not be the focus of the title.**

We modified the title to “Structural Basis for the Activation of Acid Ceramidase”.

- 25. Results, p6: The % sequence identity of naked mole rat and common minke whale to human acid ceramidase should be included.**

Done

- 26. Results, p7: The sequence identity between nmr and cmw aCDase is quoted to describe the similarity of the overall structures. The RMSD between these structures should also be included here.**

Done

27. **Panel C of Figure 1 is unnecessary: the differences between these two structures is very minimal and doesn't seem to have any importance biologically so this panel should be removed.**

Panel is now moved to Supplementary Figure 1.

28. **Figure 2 is not very informative and should be removed. The discussion of the comparison with these other homologous proteins is not very informative either. The discussion of the differences in the alpha subunit don't lead anywhere – does lacking an alpha subunit completely change the access to the active site for example?**

Figure 2 is now moved to Supplementary Figure 1.

29. **Results, p9: I think the authors mean “preceded by” not “followed by” when discussing the tight turn near residues 142 and the helix which forms the C-terminus of the alpha subunit.**

Changed “followed” to “preceded”.

30. **What is the point of the inset panel in figure 3A showing a surface representation of the helix near C143?**

The idea is to show that Cys143 is buried within the core of the protein and that the active site is blocked by helix 5 in the inactive structure. The figure was modified to show only the more relevant panel (now Fig. 2A).

31. **Supp Figure 3 is difficult to understand and it's not clear from the text (p12) what the relevance is of this conformational change. If there is some importance to this movement it needs to be more clearly explained and the figure improved to make it clear how panels A and B relate to the overall structures.**

Supp Fig. 3 now Supp. Fig 5C has been modified to more clearly depict the change in environment of Y137 upon cleavage. The main point of this figure is to show that Y137 is at the center of stabilizing the inactive straight and active bent conformation of helix 5, the bending of which allows access to the active site.

We clarified the text to “This conformational change repositions Tyr 137 relative to the active site strand β 1, from a hydrophobic environment in the proenzyme, to one where the bent conformation of helix α 5 is stabilized by hydrogen bonds between the side chain hydroxyl and backbone atoms, thus allowing access to the active site (Supp. Fig. 5C).”

- 32. Discussion, p18: Can the authors speculate as to why the neutral ceramidase would use an entirely different fold and mechanism to cleave the same substrate?**

The only other mammalian ceramidase for which a structure is known is the human neutral ceramidase (nCDase) [43]. However, nCDase is a single-pass transmembrane protein mainly expressed in epithelial cells of the intestine and colon, and thus acts at a higher pH value and on a different membrane environment than aCDase [6, 43, 44]. Moreover, nCDase acts on bile micelles, as opposed to the intra-lysosomal membrane aCDase requires. These differences may explain why their folds and mechanisms are unrelated, as nCDase depends on active site zinc ions to catalyze ceramide hydrolysis [43].

- 33. Discussion, p18: The authors state that neutral CDase possesses a larger tunnel that might accommodate different chain lengths. A Supp figure to show this would be helpful.**

See Supp. Fig. 7.

- 34. How is the highly hydrophobic surface of AC buried in the crystal packing? Is there dimer formation?**

aCDase purifies as a mixture of dimer and monomer. In the crystal, the hydrophobic surface area is buried and hidden from the solvent. This is done by packing of the hydrophobic surfaces with four different aCDase lattice symmetry mates.

- 35. Discussion, p20: The structure doesn't "explain" the molecular basis of protein inactivation in disease, it provides predictive power that can be tested in functional assays.**

Modified the sentence to clarify that it provides predictive power.

- 36. The two panels in Fig 7 are quite redundant. Panel B with labels identifying the two SMA- PME mutations would be just as useful.**

Modified the figure as requested.

Reviewer 2:

- 1. Methods section is incomplete: No details are given on the presented Western Blots in the methods section or figure legends (Figs. 3 and 6). Importantly, it is not stated which antibody was used and against which part of the protein it was directed. It is also not expressively stated which species is shown (Fig. 3C: human, naked mole rat, common minkle whale?). Details regarding the expected molecular weight vs. the actual size should be discussed as well. In Figure 5B appropriate statistics are missing. An increase of the n number (currently n=3) is required. The same applies to Figure 6C.**

The gels presented are coomassie-stained SDS-PAGE after purification of the enzymes. No antibodies were used. The assays were re-done and the n number was increased to 8.

- 2. The liposomal acid ceramidase assay employed in this study seems to be based on the assay described by Alayoubi et al., although this is not cited. The “source” of the protocol should be mentioned. Unlike previously reported acid ceramidase assays, the authors employ a fatty acid fluorimetric assay kit to quantify ceramide hydrolysis. Please validate this assay kit.**

See validation report. (Reviewer Figure 1). A reference for the source of the liposome-based assay protocol was added in the methods section.

- 3. Acid ceramidase also catalyzes the reverse reaction (acetylation of sphingosine to ceramide) at slightly higher pH (Okino et al. 2003). The authors should discuss the identified structure also with regard to this mechanism.**

In the family of Ntn hydrolases, of which acid ceramidase is part, certain enzymes carry out reactions in both directions, such as the penicillin acylase [Duggleby HJ, Tolley SP, Hill CP, Dodson EJ, Dodson G, Moody PC. Nature. 1995 Jan 19;373(6511):264-8. PMID 7816145]. In that case however, hydrolysis is favored in alkaline conditions, and acylation, at acidic pH. For acid ceramidase, because the pH optima for the two reactions are so close (4.5 and 5.5), it is difficult to rationalize this dependence. One possibility is enhanced binding of the reaction product, free fatty acid, to the enzyme above pH 5. The active site is positively charged at both pH values [Reviewer Figure 3, A and B], whereas the pKa of fatty acids is around 4.8, so at higher pH, a negatively charged fatty acid could bind with higher affinity and act as substrate.

- 4. The title seems vague and does not describe the content properly.**

Modified the title to “Structural Basis for the Activation of Acid Ceramidase”

- 5. Farber Disease is abbreviated “FB” throughout the manuscript. The more common “FD” abbreviation would be preferable.**

Changed “FB” to “FD” throughout the text.

6. Page 7, line 7: The mentioned red arrow is in Figure 1B, not 1A.

Fixed

Reviewer 3:

1. How does the proposed autocatalytic mechanism for aCDase compare to other Ntn-hydrolase members with regards to the critical residues Arg159 and Thr141/water molecule?

Deprotonation of the active site Cys (or Ser, Thr) by a nearby water molecule, thereby increasing the nucleophilicity of this residue for self-proteolysis, was also proposed for other Ntn hydrolases. In the structure of the cephalosporin acylase (CA) precursor [45], a bound water molecule is present in the same position as in the aCDase precursor [Supp. Fig. 9] and is hydrogen-bonded to corresponding residues, with one difference: in CA, one of the proton acceptors is the backbone carbonyl oxygen of the residue corresponding to Thr141, whereas in aCDase that role is played by the side chain of Thr141. This is because the residue in position 142 is a glycine in CA, resulting in a different backbone angle. The importance of this water molecule was confirmed in CA by a mutation that distorts the active site and results in loss of the water molecule from the structure; that mutation disabled self-proteolysis [46]. A conserved water molecule was also proposed to act as base in the self-cleavage of proteasome beta subunits [47]. However, this mechanism does not extend to all Ntn hydrolases; for instance, an aspartate acts as base in glycosylasparaginase self-proteolysis [48].

Here, we introduced the T141V mutant into human aCDase and measured its effect on the autocleavage rate. Mutation of Thr 141 significantly reduced the autocleavage activity of aCDase, down to 40% of the wild-type's rate (Supp. Figure 5A), although the mutant was almost fully cleaved after a 65h incubation at acidic pH (Figure 2C). Since W1 potentially hydrogen bonds with 3 other atoms in the protein (the side chain of Asn 320, the backbone amine of Glu 225, and the side chain of Asn 320, and another water molecule), mutation of Thr 141 probably only partially dislodges W1 to inhibit autocleavage.

Reviewer Figure 1 | Validation of the liposomal assay. The measured fluorescence was assessed in various samples to determine the contributions of the various components of the assay. AC=Acid Ceramidase; SapD=Saposin D; Lip=Liposomes; Det=Detergent (Triton X-100). The activities of the acid ceramidase in the presence of SapD presented in Figures 4B, 6C, and Supplementary Fig. 6B were corrected according to the background presented in the sample containing liposomes, detergent, and SapD; the activity of acid ceramidase in the absence of SapD (Supplementary Fig. 6B) was corrected according to the background presented in the

sample containing liposomes and detergent. Data for the validation assay are the means and standard deviations of six replicates.

Reviewer Figure 2| Surface view of ceramide-bound aCDase colored according to the B-factor values of each residue. Low temperature factors are colored in blue, while high temperature factors, indicative of relative higher mobility, are shown in red. Residues within the active site have relatively low mobility.

Reviewer Figure 3 | Electrostatic potential of acid ceramidase contoured at ± 5 kT/e, calculated by APBS at pH 4.5 (**A**) or 5.5 (**B**). Blue regions represent positively-charged residues, white is uncharged, and red is negatively-charged.

Reviewers' comments:

Reviewer #1 (Remarks to the Author):

Reviewer 1

The changes made by Gebai, Gorelik et al have greatly improved the manuscript. However, there remain some points I would like addressed.

In the authors response to point 6 the authors state that they can't address questions related to pH dependency as the catalytic mutations abolish autocleavage. Perhaps they have misunderstood my point. There are a number of catalytic mutations they have made that are correctly cleaved otherwise Fig 4B would be meaningless. My point was that the other family members with related active sites and related mechanisms do not seem to be optimised for acidic pH and perhaps their structure sheds light on this. I accept however that in this case pH studies may fall outside the scope of the current manuscript.

In response to point 10 the authors have produced activity assays showing that there is indeed a reasonable level of background activity in their assay in the absence of acid ceramidase. Although I accept that their activity data are acceptable to retain in the manuscript I think it is important that this observation (Reviewer Figure 1) be included as supplementary material and the correction it requires be included in the methods.

The response to point 11 makes it clear that the residues near the active site were fixed during ceramide docking to eliminate clashes. This should be clearly stated in the methods. My point here was that there are several examples of how active site residues (despite low B-factors) can move upon substrate binding and in fact require conformational changes to carry out their catalytic mechanism. The risk with fixing these sidechains for the docking is that it may force an unrealistic conformation upon both the enzyme active site and the substrate. However, without an actual structure with a substrate or product I accept this is likely to be a reasonable approximation based on the current data.

In response to point 13 the authors have included an additional Supp figure 8 to show that ceramides with bulky headgroups can't bind the active site. Although I accept this conclusion may be correct the figure provided is not at all clear regarding what these headgroups would clash with. The reason why I think this is an important point and why a convincing figure should be provided is that there is published data to suggest that substrates with sugar headgroups inhibit acid ceramidase (Pizzirani et al) and sphingomyelin inhibits the reverse activity (Okino et al). If the authors are really convinced that there is not sufficient plasticity in the active site pocket for these headgroups to fit then they can leave the text as is but provide a more convincing Supp Fig 8. However, I would caution that it might be worth having a measured discussion of this in light of the previously published work. Ideally lipids such as sphingomyelin and glucosylceramide would have been used in the activity assays to test their statement that they can't bind and to validate/challenge the previously published data.

Refs:

Pizzirani, D. et al. Discovery of a new class of highly potent inhibitors of acid ceramidase: synthesis and structure-activity relationship (SAR). *J. Med. Chem.* 56, 3518-3530, (2013).

Okino, N. et al. The reverse activity of human acid ceramidase. *J Biol Chem* 278, 29948-29953, (2003).

The activity data for active site mutations shown in Fig 4B are not really discussed in the paper. Only R333 is mentioned in reference to this panel and only in comparison to its role in autoprolysis. Doesn't it seem odd that the mutations R159Q and R333Q have a greater effect on activity than N320A when based on the mechanism shown in Fig 5, N320 as critical while the

arginines are not? Perhaps the authors can add a comment to the results and/or discussion regarding what they think is going on here.

Although the new schematics for the autoproteolysis and catalytic mechanism are helpful for understanding the roles of some key residues I suspect that once these are resized for publication they will be unreadable and therefore useless. For example, Fig 2D can be read as a full-page landscape panel but would be challenging much smaller. Perhaps in order to maintain the best chance of these being legible and helpful for readers these could be moved to Supplementary where they can be maintained at full size?

In relation to this point, neither of the schematics carry the reaction through to the endpoint and show the final resolved form that is visible in the structures. Perhaps an additional panel for each showing how these reactions are resolved would aid readers for whom enzyme mechanisms are a challenge.

Throughout the new text the Supp Figures seem to be referenced in a fairly arbitrary order. The Supp material should be re-ordered to match the order it is referenced in the text.

On page 10, line 219 the reference to Supp Fig 5A should be to 6A.

Table 3 doesn't need to be in the main text, this should be moved to Supplementary material.

In the Methods section there is now a description of SapD expression and purification which is helpful. However, both murine and human SapD production are described but it's not clear where the two different versions were used. The methods section may need expanding if SapD from different species were used for different experiments.

Supplementary Figure 1 seems to have some odd artefacts around panel B and a random Active Site label. Also, Supp Fig 1 panels D and E are very faint making them hard to interpret. Was there a reason for this or is it a mistake?

Reviewer #2 (Remarks to the Author):

The authors addressed all issues raised in my review.

Reviewer 1

- 1. In the authors response to point 6 the authors state that they can't address questions related to pH dependency as the catalytic mutations abolish autocleavage. Perhaps they have misunderstood my point. There are a number of catalytic mutations they have made that are correctly cleaved otherwise Fig 4B would be meaningless. My point was that the other family members with related active sites and related mechanisms do not seem to be optimised for acidic pH and perhaps their structure sheds light on this. I accept however that in this case pH studies may fall outside the scope of the current manuscript.**

When comparing the structure of acid ceramidase with that of related Ntn hydrolases, it is not straightforward to identify elements of the active site that are optimized for a certain pH. For example, acid ceramidase can be compared with the penicillin V acylases / bile acid hydrolases which operate at neutral pH (PDB codes 2PJF, 2PVA, 2X1D), and with the mammalian lysosomal enzymes PLBD1 and PLBD2; these two families are its closest structural homologs. The nucleophilic residue in acid ceramidase is a cysteine, which is more easily deprotonated at acidic pH than a serine. However, the penicillin acylases use both types of residues, and the PLBD enzymes employ a serine. Asn320 and R333 (Fig. 4B) are fully conserved in all these proteins. Asp162, which likely participates in catalysis in acid ceramidase (Fig. 5B), is conserved in the penicillin acylases but is replaced by a tryptophan in PLBD1/2. Finally, Arg159 (Fig. 4B) is only conserved in the bile salt hydrolase. In summary, it would be difficult to correlate the active site arrangement with pH dependency without additional experimental studies.

- 2. In response to point 10 the authors have produced activity assays showing that there is indeed a reasonable level of background activity in their assay in the absence of acid ceramidase. Although I accept that their activity data are acceptable to retain in the manuscript I think it is important that this observation (Reviewer Figure 1) be included as supplementary material and the correction it requires be included in the methods.**

The validation assay has been added in the supplementary figures and the methods were modified accordingly.

- 3. The response to point 11 makes it clear that the residues near the active site were fixed during ceramide docking to eliminate clashes. This should be clearly stated in the methods. My point here was that there are several examples of how active site residues (despite low B-factors) can move upon substrate binding and in fact require conformational changes to carry out their catalytic mechanism. The risk with fixing these sidechains for the docking is that it may force an unrealistic conformation upon both the enzyme active site and the substrate. However, without an actual structure with a substrate or product I accept this is likely to be a reasonable approximation based on the current data.**

Fixing the active site residues led to a more stable minimization.

The following sentence is now included in the methods: "Residues near the active site were fixed to improve the stability of the energy minimization procedure using Groningen

Machine for Chemicals Simulations (GROMACS) 4.6.2 package [67] with GROMOS 96 force field [68].”

- 4. In response to point 13 the authors have included an additional Supp figure 8 to show that ceramides with bulky headgroups can't bind the active site. Although I accept this conclusion may be correct the figure provided is not at all clear regarding what these headgroups would clash with. The reason why I think this is an important point and why a convincing figure should be provided is that there is published data to suggest that substrates with sugar headgroups inhibit acid ceramidase (Pizzirani et al) and sphingomyelin inhibits the reverse activity (Okino et al). If the authors are really convinced that there is not sufficient plasticity in the active site pocket for these headgroups to fit then they can leave the text as is but provide a more convincing Supp Fig 8. However, I would caution that it might be worth having a measured discussion of this in light of the previously published work. Ideally lipids such as sphingomyelin and glucosylceramide would have been used in the activity assays to test their statement that they can't bind and to validate/challenge the previously published data.**

Refs:

Pizzirani, D. et al. Discovery of a new class of highly potent inhibitors of acid ceramidase: synthesis and structure-activity relationship (SAR). *J. Med. Chem.* 56, 3518-3530, (2013).

Okino, N. et al. The reverse activity of human acid ceramidase. *J Biol Chem* 278, 29948-29953, (2003).

Thank you for providing this information. In the study of Pizzirani et al., the reported inhibitors do indeed have bulky polar headgroups, but they have only one short hydrophobic tail, so it is not certain how deeply these molecules enter into the active site, compared to ceramide. In Okino et., the results are rather scattered and conflicting. Some lipids only inhibit the forward reaction, some only the reverse, and it seems that almost any type of lipid has some effect. We suspect these effects may be due to the lipids influencing the substrate-containing micelles in some way, and not necessarily by blocking the active site.

To take into about these previous findings we changed the text on pg. 12 from:

“The shape of the substrate binding channel appears to be specific for ceramide, as other membrane-resident lipids with bulky head groups such as sphingomyelin, phospholipids and cerebroside, would result in steric clashes (Supplementary Fig. 8).”

To:

“The shape of the substrate binding channel appears to be specific for ceramide, as other membrane-resident lipids with bulky head groups such as sphingomyelin, phospholipids and cerebroside, would result in steric clashes were they to bind in a manner similar to the modelled ceramide (Supplementary Fig. 8). However, certain lipids were reported to inhibit hydrolysis or synthesis of ceramide by aCDase [40]; therefore, the interaction of various lipids with the enzyme merits further investigation.

We also modified the figure to more clearly show the numerous clashes that result from lipid headgroups.

- 5. The activity data for active site mutations shown in Fig 4B are not really discussed in the paper. Only R333 is mentioned in reference to this panel and only in comparison to its role in autoproteolysis. Doesn't it seem odd that the mutations R159Q and R333Q have a greater effect on activity than N320A when based on the mechanism shown in Fig 5, N320 as critical while the arginines are not? Perhaps the authors can add a comment to the results and/or discussion regarding what they think is going on here.**

We expanded the discussion on the mutational analysis by including the following paragraph on pg. 13:

“To assess the validity of the ceramide modelling, we made several active site mutations (Figure 4B). R333Q impaired substrate hydrolysis almost as much as C143A highlighting its importance in both orienting Cys 143 and potential interaction with the substrate; the same mutation had a weaker effect on autocleavage (Figure 2C). Likewise, D162N had a strong effect due to its potential role in stabilizing the N-terminal positive charge. R159Q also had a significant impact on activity – although Arg 159 does not hydrogen bond with Cys 143 in the active state, it is still likely important in positioning the nucleophile through Van der Waal interactions. Finally, N320A had the weakest effect presumably because its posited role as the oxyanion hole could also be offered by Glu 225 and/or the N-terminus (Figure 4D).”

- 6. Although the new schematics for the autoproteolysis and catalytic mechanism are helpful for understanding the roles of some key residues I suspect that once these are resized for publication they will be unreadable and therefore useless. For example, Fig 2D can be read as a full-page landscape panel but would be challenging much smaller. Perhaps in order to maintain the best chance of these being legible and helpful for readers these could be moved to Supplementary where they can be maintained at full size?**

The mechanism schematics were moved to the Supplementary Figures.

- 7. In relation to this point, neither of the schematics carry the reaction through to the endpoint and show the final resolved form that is visible in the structures. Perhaps an additional panel for each showing how these reactions are resolved would aid readers for whom enzyme mechanisms are a challenge.**

An additional step was added at the end of each mechanism to show how the reactions are resolved.

- 8. Throughout the new text the Supp Figures seem to be referenced in a fairly arbitrary order. The Supp material should be re-ordered to match the order it is referenced in the text.**

The supplementary figures have been reorganized.

- 9. On page 10, line 219 the reference to Supp Fig 5A should be to 6A.**

The Supplementary Figures in the text have been renumbered and the reference has been fixed.

10. Table 3 doesn't need to be in the main text, this should be moved to Supplementary material.

Table 3 was moved to Supplementary Table 1

11. In the Methods section there is now a description of SapD expression and purification which is helpful. However, both murine and human SapD production are described but it's not clear where the two different versions were used. The methods section may need expanding if SapD from different species were used for different experiments.

Only human SapD was used in the final results presented here. The purification protocol of murine SapD was added by mistake. Therefore, the latter was removed.

12. Supplementary Figure 1 seems to have some odd artefacts around panel B and a random Active Site label. Also, Supp Fig 1 panels D and E are very faint making them hard to interpret. Was there a reason for this or is it a mistake?

Thank you for pointing this out. There seems to have been a conversion error when the file was uploaded causing the artefact to appear around panel B and making panels 1 and D very faint. We modified the figure to ensure no other artefacts would appear.

The active site label in panel B should be pointing towards the location of the active C143 residue, to show the movement of the loop with respect to the active site's location.

REVIEWERS' COMMENTS:

Reviewer #1 (Remarks to the Author):

Thank you to the authors for making the changes to the latest version of their manuscript - these satisfy my comments. My only point is that Supp Fig 1 still seems to have the artefacts that the authors thought they had corrected.

REVIEWERS' COMMENTS:

Reviewer #1 (Remarks to the Author):

1. Thank you to the authors for making the changes to the latest version of their manuscript - these satisfy my comments. My only point is that Supp Fig 1 still seems to have the artefacts that the authors thought they had corrected.

The artefact appeared again from a conversion error on the *Nature Comm.* submission page. We have converted the Supplementary Figures file to a PDF file ourselves and submitted it to prevent this artefact from appearing.